# Enhancing hydrovoltaic power generation through heat conduction effects

Lianhui Li[1,5], Sijia Feng[1,5], Yuanyuan Bai[1], Xianqing Yang[1], Mengyuan Liu[1], Mingming Hao[1], Shuqi Wang[1], Yue Wu[1], Fuqin Sun[1], Zheng Liu [2] & Ting Zhang [1,3,4 ✉]

Restricted ambient temperature and slow heat replenishment in the phase transition of water molecules severely limit the performance of the evaporation-induced hydrovoltaic generators. Here we demonstrate a heat conduction effect enhanced hydrovoltaic power generator by integrating a flexible ionic thermoelectric gelatin material with a porous dual-size $Al_2O_3$ hydrovoltaic generator. In the hybrid heat conduction effect enhanced hydrovoltaic power generator, the ionic thermoelectric gelatin material can effectively improve the heat conduction between hydrovoltaic generator and near environment, thus increasing the water evaporation rate to improve the output voltage. Synergistically, hydrovoltaic generator part with continuous water evaporation can induce a constant temperature difference for the thermoelectric generator. Moreover, the system can efficiently achieve solar-to-thermal conversion to raise the temperature difference, accompanied by a stable open circuit voltage of 6.4 V for the hydrovoltaic generator module, the highest value yet.

[1] i-Lab, Key Laboratory of multifunctional nanomaterials and smart systems, Suzhou Institute of Nano-Tech and Nano-Bionics (SINANO), Chinese Academy of Sciences (CAS), 398 Ruoshui Road, 215123 Suzhou, P. R. China. [2] School of Materials Science and Engineering, Nanyang Technological University, Singapore, Singapore. [3] Center for Excellence in Brain Science and Intelligence Technology, Chinese Academy of Sciences (CAS), 320 Yueyang Rod, 200031 Shanghai, China. [4] Gusu laboratory of materials, 388 Ruoshui Road, 215123 Suzhou, P. R. China. [5] These authors contributed equally: Lianhui Li, Sijia Feng. ✉email: tzhang2009@sinano.ac.cn

**N**atural water is the most abundant resource, covering over 70% of the Earth's surface, and evaporation is an important part of the water cycle[1,2]. Due to the spontaneity and universality of water evaporation, hydrovoltaic generators (HGs) driven by water evaporation have been extensively developed in recent years. In particular, it was demonstrated in 2017 that water evaporation from functionalized porous carbon black films can reliably generate sustained voltages and currents of up to 1 V and 100 nA under ambient conditions[3]. Subsequently, a series of water-evaporation-driven HGs focusing on enhancing solid–liquid interface interactions between water molecules and functionalized nanochannels were reported[4–8]. However, restricted ambient temperature and slow heat replenishment limit the rate of water evaporation, resulting in limited performance of the HGs. Especially, the thermal gradients induced by the energy consumption during the phase transition of water molecules in evaporation have long been ignored but imply precious opportunities for environmental power harvesting.

Flexible thermoelectric materials may open up possibilities to break through these limitations of HGs. The conformability and heat conduction characteristics of flexible thermoelectric materials provide the basis for the combination with flexible HG to improve the performance, but the heat conduction matching and interface matching also remain challenges. In terms of materials, conductive polymers[9–12] and organic/inorganic hybrids[13–15] are two main kinds of flexible thermoelectric materials, which have received great attention because of their capability to convert heat into electricity directly by conformably attaching them onto curved heat sources, like human body, etc[16–19]. Current researches on flexible thermoelectric materials mainly focus on the fabrication of high-performance materials and devices[20,21], and a series of high-performance thermoelectric generators (TGs) have been developed recently[19,22–24]. However, the heat losses in the flexible thermoelectric materials caused by heat conduction and the sparse available heat gradient sources are actually serious obstacles for the development of TGs.

Here, we designed and fabricated a heat conduction effect enhanced hydrovoltaic power generator (HCEHG) by rationally integrating flexible ionic thermoelectric (i-TE) gelatin on the back of a porous dual-size $Al_2O_3$ ($d$-$Al_2O_3$) constructed a HG as heat conduction layer to improve evaporation electricity output and maintain a sustained thermoelectric conversion without any special environmental requirements. In the generator, the i-TE material can improve the heat conduction between the $d$-$Al_2O_3$ film and the near environment, thus increasing the evaporation rate of water to improve the performance of the HG to 4.0 V at 294.6 K and 30% RH. Synergistically, the HG module with continuous water evaporation can provide a constant temperature difference of ~2.0 K for the thermoelectric generator. Impressively, the black surface of the thermoelectric module can efficiently convert solar irradiation to heat, and the temperature difference reaches 4 K, accompanied by a stable open-circuit voltage ($V_{hoc}$) of 6.4 V for the hydrovoltaic part under 1 standard sun, which is the largest value for the reported hydrovoltaic generators to our knowledge. Moreover, the generated electricity can not only be stored in commercial supercapacitors but also directly drive electronic devices, such as digital calculators, or be used as an energy supply platform for wearable devices.

## Results
### Composition and working mechanism of the flexible HCEHG.
Figure 1 schematically briefly illustrates the composition and working mechanism of the flexible HCEHG. The HCEHG consists of a porous $d$-$Al_2O_3$-constructed hydrovoltaic generator, a gelatin-$K_4[Fe(CN)_6]/K_3[Fe(CN)_6]$ TG and a black encapsulation layer (BEL), which are synergistically combined to achieve a higher environmental thermopower harvesting efficiency without trade-offs. Spontaneous water evaporation accompanied by heat absorption drives the water to flow through the $d$-$Al_2O_3$-formed electrical double layer (EDL) nanochannels, generating sustainable electricity and a low-temperature region. Thus, a natural thermal gradient is constructed between the low-temperature region and the surrounding environment, which sets the stage for the $[Fe(CN)_6]^{4-}/[Fe(CN)_6]^{3-}$ redox couple in the TG module to operate. At the same time, TE gelatin with a better thermal conductivity of 0.463 W m$^{-1}$K$^{-1}$ than air of 0.0267 W m$^{-1}$K$^{-1}$ can transfer heat from the ambient environment or from photothermal conversion to the $d$-$Al_2O_3$ hydrovoltaic layer to improve the performance of HG. When exposed to sunlight, the black encapsulation layer can efficiently achieve photothermal conversion, raising the temperature gradient on the TG module and the temperature of the system to further improve the power generation performance.

**Porous $d$-$Al_2O_3$ HG**. The fabrication of the porous $d$-$Al_2O_3$ HG is schematically illustrated in Supplementary Fig. 1, and the details are given in the Methods section. Chemically inert conductive carbon paste was printed on a 200-μm-thick PET substrate by screen printing to obtain a PET substrate with electrodes. Then, an ethanol suspension of $Al_2O_3$ composite nanoparticles with diameters of 200 and 20 nm (mass ratio of 50:1) was scrape coated on the above PET substrate to form a HG. Attributed to capillary-driven self-assembly in the drying process of the suspension, a solid ~60-μm-thick porous $d$-$Al_2O_3$ film can adhere tightly to the substrate (shown in Supplementary Fig. 2).

A long real-time open-circuit voltage test was employed to verify the power generation performance of the obtained $d$-$Al_2O_3$ HG. As shown in Fig. 2a, a $V_{oc}$ of ~4 V was achieved when the bottom of the device was immersed in distilled water under natural room conditions of 296.0 K and 22.3% RH. To explain the working mechanism of the HG, we investigated the X-ray photoelectron spectroscopy (XPS) of C elements of the top and bottom electrodes (the $Al_2O_3$ covered part) of a device after continuous working for 24 h. As shown in Supplementary Fig. 3, nearly identical spectra rule out the electricity generated from redox reaction of chemically inert carbon electrodes. Further, the film morphology and $d$-$Al_2O_3$ nanoparticle zeta potential were performed. As displayed in Fig. 2b and Supplementary Fig. 4, the $d$-$Al_2O_3$ film consists of porous nanochannels, and the zeta potential in pH-neutral solution is as high as +48 mV, which is consistent with the reported streaming potential mechanism. Therefore, we propose the following mechanism: evaporation occurs on the entire surface of the $d$-$Al_2O_3$ film, driving water upward through the nanochannels, and the evaporation of water ($Q_{eva}$) reaches a balance with the capillary seepage flux ($Q_{cap}$) (Fig. 2c). The positively charged nanochannels repel positively charged ions (H$^+$) in the evaporation-driven water flow but allow the negatively charged hydroxyl ions (OH$^-$) to pass through, inducing a streaming potential and charge accumulation along the flow that forms an electric field (Fig. 2c)[8,25]. The accumulated charge forms a diffusion current ($I_{diffuse}$) in the opposite direction of the streaming current ($I_{sc}$) due to the coulomb action. At steady state, the $I_{sc}$ and $I_{diffuse}$ will reach dynamic equilibrium, i.e. $|I_{sc}| = |I_{diffuse}|$[26–28]. As a result, the device reaches a state where the stable open circuit voltage ($V_{hoc}$) is numerically stable and the top electrode is negative. Repeatedly switching the connection mode between the device and the voltmeter, the absolute voltage value is constant and the direction of signal changes correspondingly, further confirming the proposed working mechanism (Supplementary Fig. 5).

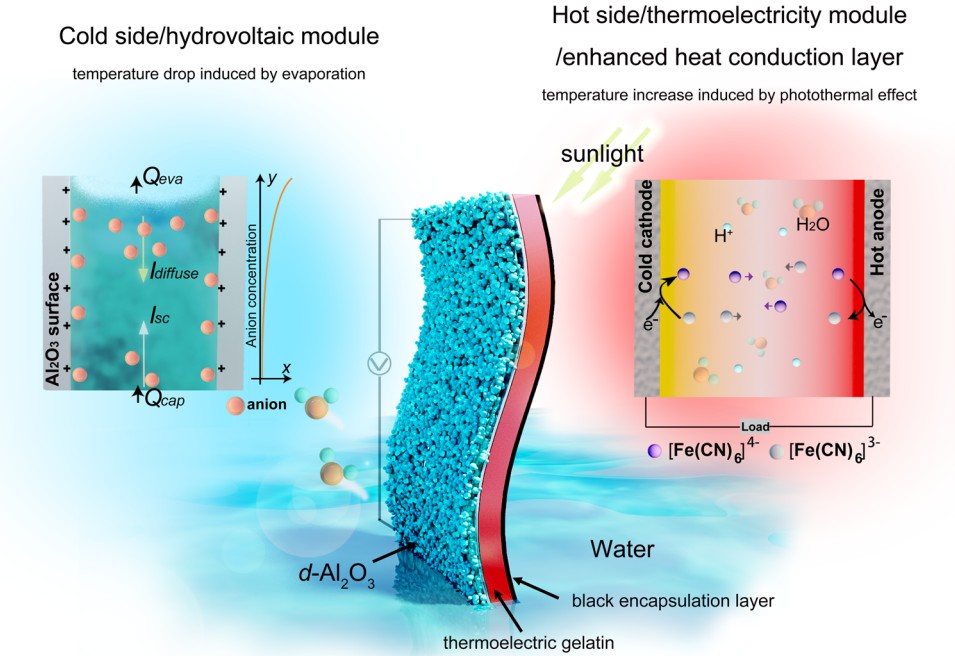

**Fig. 1 Schematic diagram of the HCEHG.** Schematic diagram of the structure and mechanism of the HCEHG for sustained evaporation electricity output and thermoelectric conversation without any special environmental requirements.

We changed the mass ratio of $m_{200}$: $m_{20}$ from 1:0 to 50:1 ($m_{200}$ and $m_{20}$ are the mass of 200 and 20 nm $Al_2O_3$ respectively.), and the short-circuit current ($I_{hsc}$) of the HGs increased from 1.05 µA to 1.34 µA while the $V_{hoc}$ remains almost constant (Fig. 2d). That comes down to the size effect brought by $d$-$Al_2O_3$ nanoparticles: small (~20 nm) $Al_2O_3$ particles can fill into the gap formed by large (200 nm) $Al_2O_3$ particles and generate smaller nano channels to improve its selectivity and device performance (Fig. 2c). Compared to previously reported works[3,25,29–31], the water source used here is readily accessible distilled water, and the voltage generated by HGs using distilled water is almost the same as that generated using deionized water (Supplementary Fig. 6). However, excessive small $Al_2O_3$ particles ($m_{20}$:$m_{200}$ ≥ 1:5) will dramatically increase the flow resistance of the channel and introduce a large number of cracks to reduce the performance of the device (Fig. 2d and Supplementary Fig. 7).

The fabricated flexible porous $d$-$Al_2O_3$ HG can withstand a large range of bending deformations of 0 to 180° without significant attenuation in the power generation performance (Fig. 2e and Supplementary Fig. 8). That is because the capillary-driven self-assembly in the drying process of the suspension allows porous $d$-$Al_2O_3$ to bind well with flexible substrates. On the other hand, COMSOL simulations indicate that the small device thickness (substrate thickness ~200 µm) only induced 0.111% maximum strain locally in the film when bended to 180° (Supplementary Fig. 9). The generated voltage of the obtained $d$-$Al_2O_3$ HG module does not vary with the width of $d$-$Al_2O_3$ in the range of 1 to 12 cm under a constant height of 3 cm, while $I_{hsc}$ is positively correlated with the width. As presented in Supplementary Fig. 10, $I_{hsc}$ increased from 0.17 to 1.81 µA when the width changed from 1 to 12 cm at 303.2 K and 60% RH, which can be explained as a series of generators running in parallel on electrical circuits. Due to the internal resistance of the $d$-$Al_2O_3$ film and the water climbing height, the height of the device has a significant effect on both the voltage and current, and the optimal value is achieved at 3 cm (Supplementary Figs. 11 and 12). Increasing the thickness of the $d$-$Al_2O_3$ layer can also increase the current of the device. However, when it is >60 µm,

the mechanical strength of the device will be significantly reduced (Supplementary Fig. 13a–c). The influence of size on the output performance of the device provides a theoretical basis for the number and size expansion of flexible porous $d$-$Al_2O_3$ HGs. For a $10 \times 4$ cm² generator (width × height), the effective output power can reach 1.84 µW when the load resistance is 2.22 MΩ (Fig. 2f) at 303.2 K and 60% RH. Here, the output power was tested by connecting a series of load resistors with different resistances, and the circuit diagram is shown in Supplementary Fig. 14.

The evaporation rate of water can dramatically affect the performance of the flexible porous $d$-$Al_2O_3$ HG. When the air velocity increased from 0 to 0.04 to 0.4 m s⁻¹ (tested by a Fluke F923 hotwire anemometer), the $V_{hoc}$ of the generator correspondingly increased from 2.2 to 3.2 to 3.9 V (16.48 °C and 45.77% RH, Supplementary Fig. 15). Humidity and temperature are two other key factors. As expected, when keeping the ambient temperature constant at 30 °C and increasing the ambient humidity from 40% to 80% RH, $V_{hoc}$ decreased from 5.84 to 1.33 V (Supplementary Fig. 16). Increasing the ambient temperature from 293.2 to 313.2 K can also improve the output voltage from 2.93 to 4.82 V when the ambient humidity is constant at 60% RH (Supplementary Fig. 17). It is worth noting that the impact of the local temperature field on the device performance conforms to the ambient temperature field (Supplementary Fig. 18).

**i-TE gelatin TG**. An i-TE material with thermal conductivity of 0.463 W m⁻¹K⁻¹ (17 times more than air) composed of an organic gelatin matrix and an $[Fe(CN)_6]^{4-}$/$[Fe(CN)_6]^{3-}$ redox couple was employed to construct a TG with carbon fibre film (CFF) electrodes as heat conduction enhanced module of the HCEHG (Fig. 3a). The details of the fabrication process are given in the Methods section. The mechanically flexible quasi-solid-state i-TE material (Fig. 3b) and the CFF electrode with a large specific surface area (Fig. 3c) endow the i-TE material with enhanced current density and good binding with the electrode without any mechanical mismatch (Supplementary Fig. 19).

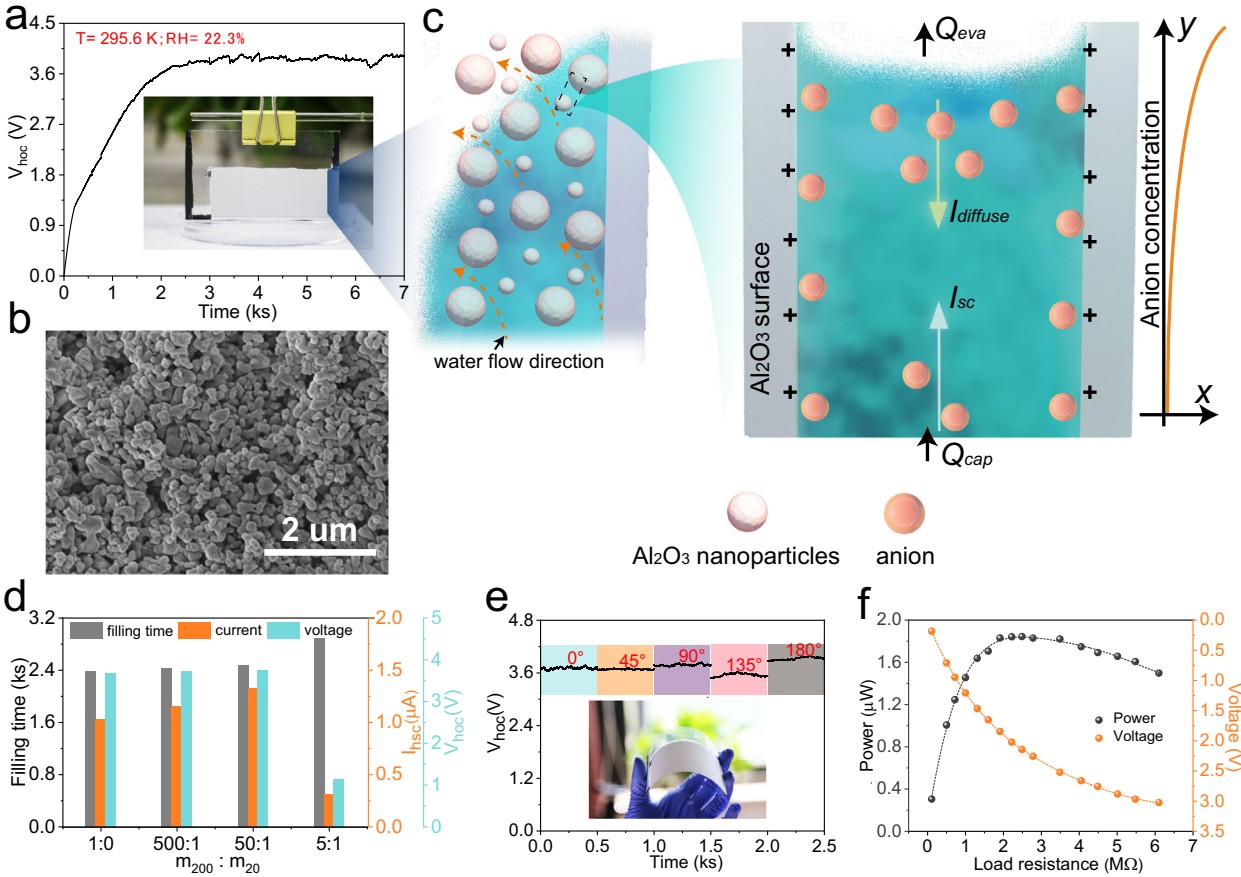

**Fig. 2 Power generation performance of the flexible porous *d*-Al₂O₃ HGs. a** Open-circuit voltage response versus time curve at an ambient temperature of 295.6 K and a humidity of 22.3% RH. Inset: typical photograph showing the working state of the generator. **b** Scanning electron microscopy (SEM) image of a porous *d*-Al₂O₃ film. **c** Schematic depiction of the working mechanism of the evaporation-driven HG. **d** Variation of filling time, $V_{hoc}$, and $I_{hsc}$ as a function of mass ratio of 200 and 20 nm Al₂O₃. **e** Measured $V_{hoc}$ of the nanogenerator in different bending states. Inset: photographs illustrating the flexibility of the porous *d*-Al₂O₃-based power generator. **f** Output voltage and power of the power generator (size: width: 10 cm, height: 3 cm) as functions of the load resistance in a room environment.

In this TG system, $[Fe(CN)_6]^{4-}$ has a lower solvation entropy, and the oxidation reaction $[Fe(CN)_6]^{4-} \rightarrow e + [Fe(CN)_6]^{3-}$ is thermodynamically favourable, which leads to the injection of electrons into the hot electrode, thus increasing the electrochemical potential (i.e., lowering the voltage), as illustrated in the schematic diagram in Fig. 3d. At the same time, the reduction reaction $[Fe(CN)_6]^{3-} + e \rightarrow [Fe(CN)_6]^{4-}$ with electrons from the electrode is thermodynamically favoured, inducing a decreased electrochemical potential (i.e., a higher voltage). The redox couple works together to achieve sustainable output power[14,24,32,33].

Then, the thermoelectric performance of the TG was evaluated by applying a controllable temperature gradient using an electrical heating plate (on the top) and a water-cooled plate (on the bottom). The test setup diagram is shown in Supplementary Fig. 20. When we maintained the cold side at a constant 293 K and increased the hot side temperature, the open-circuit voltage ($V_{toc}$) generated by the TG ($V_{toc}$) changed from 2.8 mV at $\Delta T = 0.8$ K to 36.4 mV at $\Delta T = 42.2$ K, and the Seebeck coefficient was ~0.82 mV K⁻¹ in the $\Delta T$ range of 0.8 to 42.2 K (Fig. 3e and Supplementary Fig. 21). $\Delta T$ is the temperature difference between the hot side and the cold side. In this TG system, the Seebeck coefficient is defined as refs. [14,34,35]:

$$S_e = \Delta E / \Delta T = \Delta S / nF \quad (1)$$

where $\Delta E$ is the open-circuit voltage, $\Delta T$ is the temperature difference, $n$ is the number of electrons transferred in the redox

reaction, $F$ is Faraday's constant, and $\Delta S$ is the partial molar entropy difference of the redox couple. The directionality of $V_{toc}$ was further confirmed by repeatedly switching the connection mode between the device and the voltmeter, and the results were consistent with the abovementioned results (i.e., the hot electrode is negative and the cold electrode is positive). Under the same temperature differences as above, the short-circuit current of a single $1 \times 1$ cm² (length × width) TG device can reach 2.2 mA, which is considerable for flexible quasi-solid-state gel-based TGs (Fig. 3f).

We further investigated the efficient output power of the obtained TG device under different temperature differences by connecting a series of load resistors with different resistances. The effective output power is calculated as $P = I^2 \times R_L$, where $P$ is the effective output power, $I$ is the current generated in the circuit, and $R_L$ is the load resistance. As shown in Fig. 3g, optimised output power densities of approximately 0.0142, 0.045 and 0.146 W m⁻² are achieved at load resistances of ~9 Ω under $\Delta T$ of 8.6, 23.8 and 42.2 K, respectively. The temperature-difference-induced potential signal can also remain stable under 40 cycles of periodic application-removal of $\Delta T = 8.6$ K and 23.6 K thermal stimuli (Fig. 3h), and the output power at $\Delta T = 8.6$ K with a constant load resistance of 200, 500, 1000 and 2000 Ω maintained over 75% of its initial value after over 50 min (Supplementary Fig. 22). These results indicate the high repeatability and stability of the as-fabricated flexible quasi-solid-state TG device.

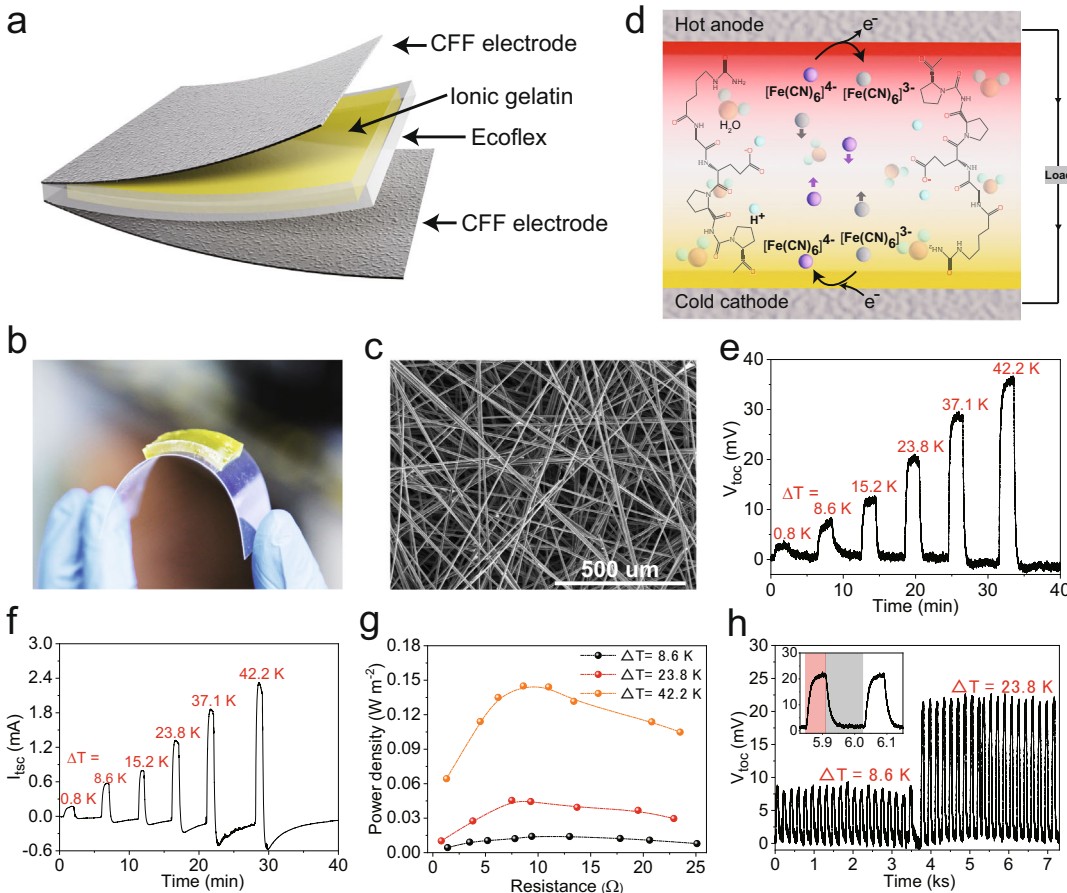

**Fig. 3 Composition and power generation performance of the flexible TG module. a** Schematic diagram of the construction of a fabricated TG. **b** Optical image of the quasi-solid-state ionic gelatin on the bent PET film showing the flexibility of the i-TE material used. **c** SEM image of the CFF film. **d** Schematic figure of the diffusion, redox reaction, and interaction of the ions in the fabricated flexible ionic gelatin under a temperature gradient. **e** Voltage generated from the as-fabricated TG at different temperature gradients. **f** Short-circuit current response versus time curve at different temperature gradients. **g** Corresponding power density at different load resistances under $\Delta T$ of 8.6, 23.8, and 42.2 K. **h** Change in voltage under a cyclic heat gradient (temperature change) from 0 to 8.6 K and 0 to 23.8 K over 20 cycles. Inset: Partial enlargement of the curve. The light red area represents the thermal is applied and the light grey area represents heat is removed.

**Power generation performance of the HCEHG**. Due to the phase transition of water from liquid to gas, the temperature of the $d$-Al$_2$O$_3$ film is lower than that of the surrounding environment, which is a promising energy source for TG devices to convert ambient heat into electricity. We rationally assembled flexible i-TE gelatin on the back of the porous $d$-Al$_2$O$_3$-constructed HG as heat conduction layer to form a HCEHG to harvest the abovementioned heat energy (Fig. 4a and Supplementary Fig. 23). As expected, the i-TE gelatin part was found to significantly improve the voltage of the HG from 3.4 V to 4.0 V as well as the effective output power from 1.84 μW to 2.36 μW at 294.6 K and 30% RH (Fig. 4b and Supplementary Fig. 24), which is attributed to the better thermal conductivity of TE gelatin leading to the transfer of more heat from the ambient environment to the $d$-Al$_2$O$_3$ film. Evidently, there is a 1 K temperature difference between the $d$-Al$_2$O$_3$ film surfaces of the HCEHG (high) and individual hydrovoltaic device (low), further confirming the contribution of the i-TE gelatin with high heat conductivity to the hydrovoltaic component (Supplementary Fig. 25). As the other part, the water-evaporation-induced temperature gradient between the $d$-Al$_2$O$_3$ film and the surface of the thermoelectric part packaging layer reaches ~2.0 K, which is certified by the vertical thermal mapping (Supplementary Fig. 26). The automatically thermal gradient drives the TG module to produce a stable voltage of ~1 mV (Fig. 4c).

**Enhancement of solar-to-thermal conversion**. Solar energy is undisputed clean and renewable energy, and solar-to-thermal conversion is a highly efficient and easy way to utilise solar energy[36–38]. Owing to the black encapsulation layer on the surface of the thermoelectric part, HCEHG can efficiently achieve solar-to-thermal conversion to raise the thermoelectric surface temperature from 290.1 K to 300.5 K at an optical density of 1 kW m$^{-2}$ (1 sun irradiation) (Fig. 4d–g). As the encapsulation layer warms up, heat is transferred through the i-TE gelatin to the $d$-Al$_2$O$_3$ layer, realising a temperature increase from 288.5 K to 296.7 K. Naturally, light will trigger a boom in the performance of the power generation system. As shown in Fig. 4h, the $V_{hoc}$ and $V_{toc}$ signals are boosted from 3.68 to 6.4 V and from 1 to 4 mV, respectively, when sunlight is applied at 290.1 K and 55% RH, which is mainly because the higher temperature accelerates the evaporation of water. Similar trends of $I_{hsc}$ and $I_{tsc}$ under sunlight can be observed in Fig. 4i. To the best of our knowledge, the performance of stably generating an open-circuit voltage of 6.4 V for a single device remarkably outperforms previously reported hydrovoltaic power generators (Fig. 4j)[39–44]. By contrast, the temperature of a standalone $d$-Al$_2$O$_3$ HG without the TG module only increases to 296 K, and $V_{hoc}$ is 1.4 V smaller than that of the HCEHG at an optical density of 1 kW m$^{-2}$ (Supplementary Fig. 27). Collectively, the as-fabricated flexible hybrid HCEHG is a synergistic strategy for efficient environmental energy harvesting.

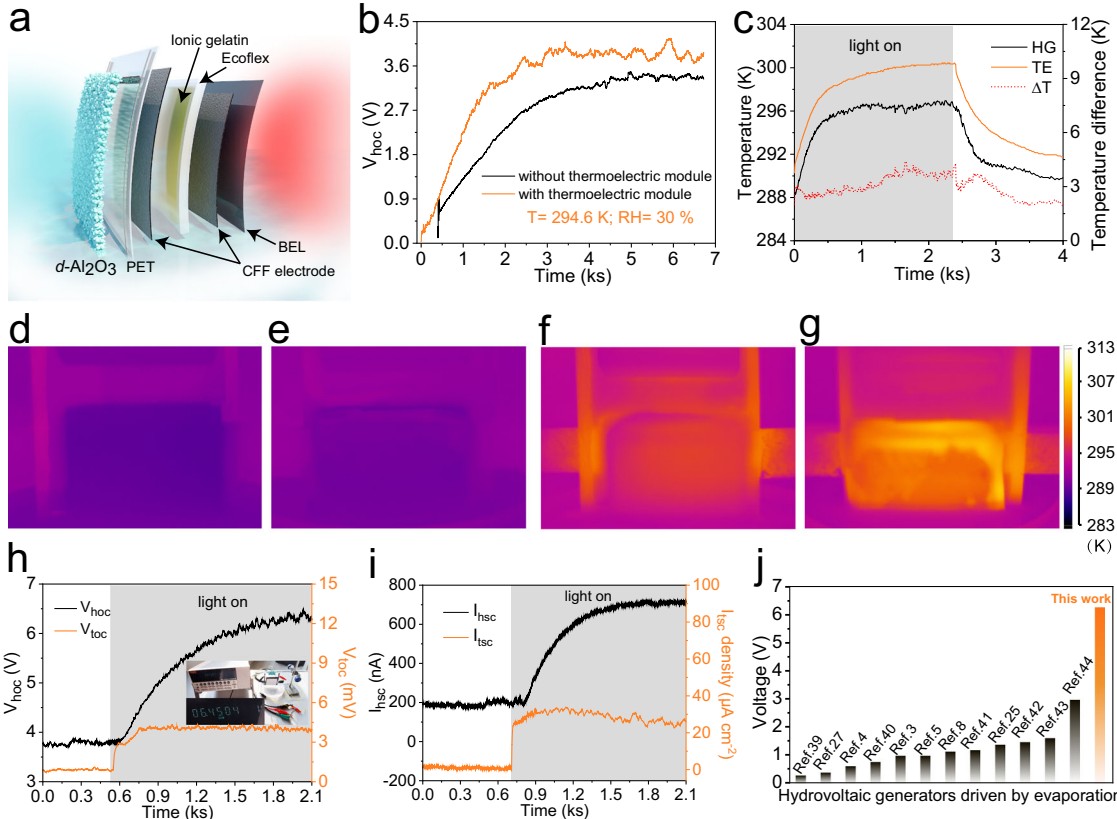

**Fig. 4 Proof-of-concept of the HCEHG. a** Schematic diagram of the construction of the HCEHG. **b** Open-circuit voltage response versus time curve of the hydrovoltaic module with and without the thermoelectric module at an ambient temperature of 294.6 K and a humidity of 30% RH. **c** Real-time temperature change and corresponding temperature difference of the surfaces of the $d$-Al$_2$O$_3$ film (i.e., HG) and TG module. The ambient temperature is ~290.4 K. Grey areas indicate that the device is illuminated. **d, e** Infrared thermal images of the $d$-Al$_2$O$_3$ film surface (**d**) and TG module surface (**e**). **f, g** Infrared thermal images of the $d$-Al$_2$O$_3$ film surface (**f**) and TG module surface (**g**) under an optical density of 1 kW m$^{-2}$. The ambient temperature is ~290.4 K. **h** Real-time $V_{hoc}$ and $V_{toc}$ changes of the hydrovoltaic module and thermoelectricity module under one sun irradiation, respectively. The size of the hydrovoltaic module is 5 × 3 cm$^2$ (width × height). Grey areas indicate that the device is illuminated. **i** Real-time $I_{hsc}$ and $I_{tsc}$ density changes of the hydrovoltaic module and thermoelectricity module under one sun irradiation, respectively. Grey areas indicate that the device is illuminated. **j** Maximum voltage generated by single HG driven by evaporation.

**Applications**. The HCEHG provides promising solutions for the power needed for digital devices, energy storage devices and wearable electronics. As presented in Fig. 5a, when we connect a digital calculator directly in series with the hydrovoltaic part of the system, the digital calculator can operate smoothly. To further prove the practical useful power levels generated by the HCEHG, the output was used to charge commercial capacitors with capacitances of 2.2, 10, 47 and 100 µF to 5.2 V within 4000 s (Fig. 5b). Moreover, these fully charged commercial capacitors can directly power red, green and blue LEDs, making the hybrid power generation system a candidate for useful, environmentally friendly, sustainable power supply platforms (Supplementary Fig. 28). In particular, the thermoelectric module can sensitively harvest the heat field generated by the human body and convert it into electrical energy. As shown in Fig. 5c, when fingers approached the device, from 5 cm to contact, the output voltage of the thermoelectric module increased from 1.5 mV to 4.1 mV. The thermoelectric module also achieved a short circuit current and power density of 241 µA and 1.1 mW m$^{-2}$ in contact with the skin, which provides a basis for generating electricity from human body heat (Supplementary Fig. 29a–b).

The stable power generation and flexibility enable our HCEHG to serve as a wearable electronic device power platform to construct a wearable power-sensor system. We designed a wearable power-sensor system to detect physiological signals by integrating the HCEHG, a flexible CNT pressure sensor and a Bluetooth device, as illustrated in Fig. 5d. Here, the superabsorbent hydrogel was chosen as the water source[4]. In this wearable system, the fabricated device works as a power supply to drive the flexible sensor, and then, the signal is sent to the mobile phone for graphical display and digital analysis of personal health. Figure 5e presents real-time pulse detection by attaching the HCEHG-driven sensor to the carotid artery, and the visualised pulse indicates that the pulse frequency is approximately 80 times per min. Moreover, the wearable power-sensor system can also recognise large-scale joint bending motions. As shown in Supplementary Fig. 30, when the neck is quickly bent from an upright position to 15°, 30°, 45° and 60° and then holding for 10 s, the real-time current will change correspondingly. The above results demonstrated the wearable power-sensor system can sensitively monitor physiological signals, further confirming its practicability.

## Discussion

In summary, we fabricated a HCEHG by rationally assembling flexible i-TE gelatin on the back of a porous $d$-Al$_2$O$_3$ constructed HG to break through the existing restricted ambient temperature and slow heat replenishment limit of HG. In the HCEHG, the thermoelectric device can improve the heat conduction between the $d$-Al$_2$O$_3$ film and the near environment, thus increasing the

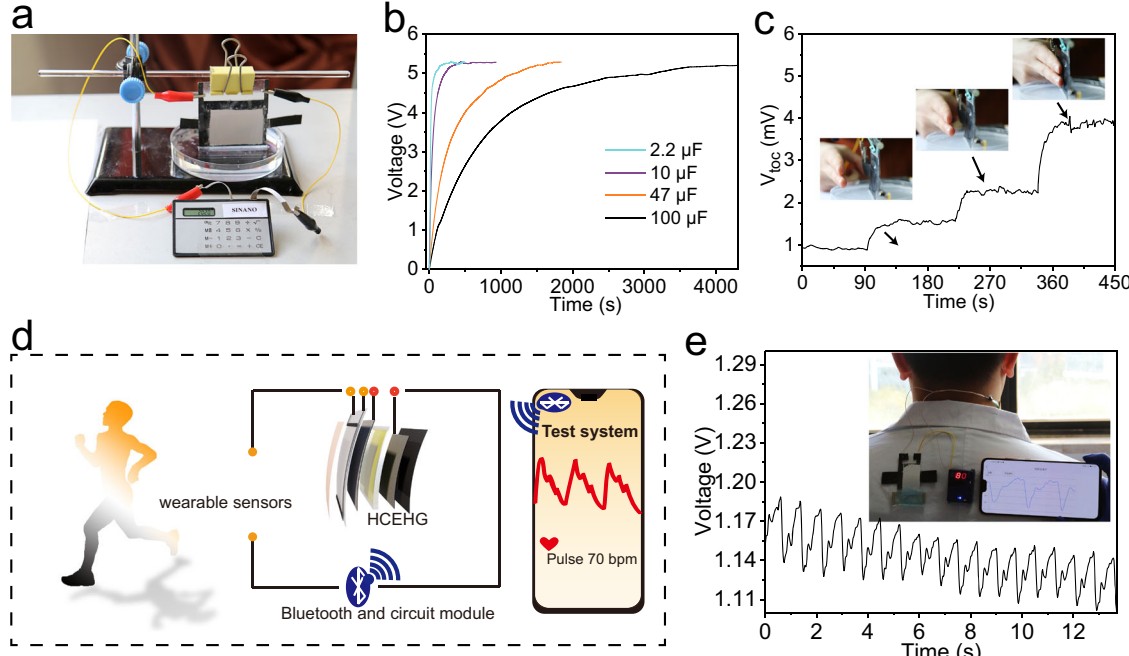

**Fig. 5 Applications of the HCEHG. a** Photograph of a digital calculator driven by the HCEHG. **b** Real-time voltage change of commercial capacitors when charged by the $d$-Al$_2$O$_3$ HG part of the HG under sunlight. **c** Real-time voltage change of the thermoelectric module when index and middle fingers approach it, from 5 cm to 2.5 cm to 0 cm. **d** Schematic diagram of a wearable power-sensor system including the HCEHG, a flexible CNT pressure sensor and a Bluetooth device. **e** Real-time pulse curve detected by attaching the HG-driven sensor to the carotid artery. Inset: photograph showing the testing process and graphical display by a mobile phone.

evaporation rate of water to improve the performance of the hydrovoltaic generator from 3.4 V to 4.0 V at 294.6 K and 30% RH. Besides the dual-size Al$_2$O$_3$ nanoparticle design can effectively achieve the enhancement of selectivity improvement for charged ions in water. Synergistically, the HG with continuous water evaporation can induce a constant temperature difference of approximately 2.0 K for the thermoelectric generator. Moreover, the black surface can efficiently achieve solar-to-thermal conversion to raise the thermoelectric surface temperature from 290.1 K to 300.5 K at an optical density of 1 kW m$^{-2}$ (1 sun), and the temperature difference will reach 4 K, accompanied by a stable $V_{hoc}$ of 6.4 V for the hydrovoltaic part, the highest value yet. In addition, the generated electricity can not only be stored in commercial supercapacitors but also directly drive electronic devices, such as digital calculators, or be used as an energy supply platform for wearable devices. The flexible HCEHG creatively uses the most common thermal gradient induced by water evaporation for thermoelectricity while achieving performance improvement of the HGs, providing a promising method for the synergistic integration of environmental energy sources (thermal, solar, etc.) to generate electricity.

## Methods

**Materials**. Aluminium oxide (200 nm, 99.99%, $M_w = 101.96$, α-phase, Macklin, China); aluminium oxide (20 nm, 99%, $M_w = 101.96$, γ-phase, Titan, China); gelatin (medicinal grade, adhesive strength ~240 g Bloom, Aladdin, China); potassium ferrocyanide trihydrate (99.0%, $M_w = 422.39$, Aladdin, China); potassium ferricyanide (99.95%, $M_w = 329.25$, Aladdin, China); conductive carbon paste (Leanstar, Soochow, China). Superabsorbent hydrogel purchased from Shenyang Cornerstone Shuanglong Chemical Co. Ltd.

**Fabrication of the HG**. Polyethylene terephthalate (PET) films were cleaned in ethanol with ultrasonication and then dried in an 80 °C oven. Conductive carbon paste was coated on PET with two L-shaped structures as electrodes. The distance between the top and bottom electrodes was 3 cm, and the line width of the electrodes was 5 mm. A mixture of dual-size Al$_2$O$_3$ particles with particle sizes of 200 nm and 20 nm (mass ratio of 50:1) was ultrasonically dispersed at 450 W for

3 h. After ultrasonication, the dispersion was settled for 12 h to obtain a $d$-Al$_2$O$_3$ slurry. Then, the $d$-Al$_2$O$_3$ slurry was evenly scrape coated on the PET substrate with electrodes. After the ethanol was evaporated, a porous $d$-Al$_2$O$_3$ functional layer with abundant nanometre channels was obtained.

**Fabrication of the TGs**. Three grams of gelatin and 8 ml of deionized water were fixed at 80 °C for 0.5 h. Then, 1.3516 × $g$ K$_4$[Fe(CN)$_6$]·3H$_2$O and 1.0536 g K$_3$[Fe(CN)$_6$] were added into the mixture to obtain homogenous thermoelectric materials after stirring at a constant temperature of 60 °C for 3 h. The thermoelectric generator was fabricated as the sandwich structure CFF|i-TE|CFF. The specific packaging method was as follows: The CFF electrode was combined with an Ecoflex mould (with a cavity of 1 × 1 × 0.27 cm$^3$) by a black encapsulation layer. After filling the unsolidified thermoelectric material into the mould, the upper electrode was encapsulated on the mould with another black encapsulation layer.

**Fabrication of the HCEHG**. The CFF and Ecoflex mould were bonded to the back of a carbon electrode/PET substrate with double-sided adhesive. After the i-TE material was filled into the mould, the other CFF electrode was sealed with a black encapsulation layer. Then, a functional layer of porous $d$-Al$_2$O$_3$ was obtained on front of the carbon electrode/PET substrate by scrape coating the $d$-Al$_2$O$_3$ slurry.

**Characterisation**. The Fluke TI400 infrared thermal imager was used to detect the temperature and infrared thermal images. A digital camera (Canon EOS 70D) was used to take the optical photographs and videos in this study. The morphology and element analyses of the samples were examined by scanning electron microscopy (SEM, JSM-7001F). X-ray photoelectron spectroscopy (XPS) data were recorded on an ESCALAB 250 photoelectron spectrometer (Thermo-Fisher Scientific) with Al Kα radiation (1486.6 eV). Thermal conductivity of the i-TE material is measured by TC3000 thermal conductivity metre fabricated by Xi'an Xiatech electronics Co., Ltd. The ambient temperature and humidity were controlled by a temperature and humidity cabinet manufactured by Guangzhou Wusuo Environmental Equipment Co., Ltd. The voltage and current signals were recorded in real time using a Keithley 6514 electrometer, which was controlled by a LabView-based data acquisition system.

## Data availability

The authors declare that the main data supporting the findings of this study are contained within the paper. Source data are provided with this paper, and all other relevant data are available from the corresponding author upon reasonable request. Source data are provided with this paper.

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

## Acknowledgements

The authors acknowledge funding support from National Key R&D Program of China 2017YFA0701101, 2018YFB1304700; National Natural Science Foundation of China 61801473, 62071463, 22109173; National Science Fund for Distinguished Young Scholars 62125112, and the Strategic Priority Research Program of Chinese Academy of Science under Grant XDB32050100. The authors are grateful for the technical support for Nano-X from Suzhou Institute of Nano-Bionics, Chinese Academy of Science (SINANO).

## Author contributions

L.L. and S.F. contributed equally to this work. T.Z. supervised the whole project. L.L., S.F. and T.Z. conceived and designed the experiments. L.L., S.F., Y.B., X.Y., M.L. and M.H. performed the experiments. L.L., S.F., S.W., Y.W. and F.S. analyzed the data. L.L., and S.F. wrote the original paper. Z.L. and T.Z. helped revised it. All authors discussed the results and commented on the paper.

## Competing interests

The authors declare no competing interests.
