## [Peer Review File · Nature Communications]

Reviewers' comments:

Reviewer #1 (Remarks to the Author):

The authors have successfully introduced a new HCEHG based on the combination of flexible i-TE gelatin and aluminum oxide to overcome the ambient temperature and slow heat replenishment limit. A comparably high voltage is achieved with a solid application example. However, several questions need to be taken into account to further improve the manuscript quality:

1. Why is the proposed porous Al₂O₃ coated to 60 microns thickness? What happens as the thickness in either the PET layer or the HG layer changes? Will it influence the as-proposed nanochannel steady-state reaching rate or streaming potential? The voltage doesn't vary from 2-10cm, what about <2 or >10?
2. Since the layer is considered to be automatically thermal gradient, a vertical thermal mapping should be carried out to prove the stated conclusion.
3. The theoretical principle behind has few innovations, but considering the high hydrovoltaic value, still a very valuable work. Maybe discuss more about intra-layer and interlayer transportation of the electrons and holes.
4. Also, the heat and moisture might not be evenly distributed in all the areas of the working device. For example, the heat and moisture distribution may not be as perfect as a heat plate. Will this lead to a variance in the evaporation rate as well as the h-e mobility?
5. Fig. 1 is composed of three figures: left, middle, and right. The direction of ammeters in the left and right figures should be the direction of positive charges, and their arrow directions are reversed. The current direction in the right diagram is reversed. The direction of the left picture is confusing. In my point of view, water vapor should move upward. So the left picture should be rotated 90 degrees counterclockwise. In this way, the meters in the left figure and the middle figure are connected at the upper and lower ends, which is easy to understand.
6. Fig S2 needs to add labels to mark these layers.
7. In lines 110-111, why switching the connection can prove this mechanism. This needs more explanation. In most cases, changing the positive and negative connection of the voltmeter will not change the absolute value measured. Under what circumstances will it change? What would happen if it changed?
8. The chemical equation in lines 150 and 153 needs to be modified. "FeCN₄-⁻" should be Fe(CN)₆⁴⁻. "FeCN₃-⁻" should be Fe(CN)₆³⁻. In line 303, the unit of 60 is missing.
9. In Fig 4 f-g, why is the middle area low temperature? If it's convenient, please add the optical image.
10. The comparison of Fig S14 may not be appropriate. Authors can coat the back of Al₂O₃ with black carbon film to harvest solar irradiation.
11. In line 266, it says the neck is gradually bent, but there are no gradually changed signals in Fig S17 instead of pulsed signals. A pressure sensor should detect pressure. And the pressure should be gradually changed if the neck is gradually bent. If the neck is suddenly turned to one degree and holds the position, the signal should also hold at a high position.

Reviewer #2 (Remarks to the Author):

The paper presents 'a flexible heat conduction effect enhanced hydrovoltaic power generator (HCEHG) by integrating a flexible ionic thermoelectric (i-TE) gelatin material with a porous dual-size Al₂O₃ (d-Al₂O₃) HG. The i-TE gelatin material can effectively improve the heat conduction between HG and near environment, that increases the water evaporation rate to improve the output voltage of the HPG to 4.0 V at 294.6 K and 30% RH. The HG part with continuous water evaporation can induce a constant temperature difference for the thermoelectric generator (TG). The system can further efficiently achieve solar-to-thermal conversion to raise the temperature difference, accompanied by a stable open circuit voltage of 6.4 V for the HG module.'

Though various experimental observations are performed, the idea of establishing such an evaporation induced power generator is not new. The content of the manuscript is not well organized as there are

repetition of several experimental observations (the parametric results should be included in the ESI and the main findings should be in the main manuscript). In addition, there are several fundamental aspects which the authors did not address properly and therefore cannot be recommended for publication. For example:

1) What is the reason for using dual size particles? What is the effect of dual size nanoparticle mass ratio on the capillary filling rate? In connection to this, how the hierarchical nanoparticle structures contribute to the streaming potential/current?

2) What kind of pore model is considered for explaining the streaming current mechanism? It seems (from SEM image), the pores are not well defined and randomly distributed, so how does it affect the current—is there any leakage current? There is a high chance of leakage current as only a few fractions of the pores are responsible for generation of streaming current. The streaming current mechanism cannot be explained by simply considering straight nanochannels.

3) Why is the carbon tape used in this measurement? Generally, Ag/AgCl electrodes are used for such measurements.

The capacitance of the carbon electrode in contact with the solution, will then play a role, because at these low voltages it is less likely that Faradaic reactions occur at the carbon electrodes may lead to a large voltage drop at the electrodes that may cause water electrolysis, complicating the analysis.

4) As the zeta potential of d-Al₂O₃ is relatively high (48 mV), a higher streaming current is expected, however, it is observed to be low here. On the other hand, the complete capillary filling time or voltage equilibrium time (Fig. 2a) is found to be quite high (around 41 min) which maybe due to the high channel resistance. Is it the mass ratio or dual particle size influencing the device performance?

5) When there is any bending deformation (Fig. 2d), the nanopores length would increase and that should decrease the output voltage. On the other hand, the increase in the surface area due to bending should increase the surface conduction effect that may lead to reducing the streaming current. On the contrary, the increased surface area also leads to an increase in the evaporation rate which should increase the current. It's quite surprising that there is no change in the output voltage---the author should judiciously justify all the contradictory factors.

6) The statement – 'Ihsc increased from 0.32 to 1.34 μ A when the width changed from 2 to 10 cm at 303.2 K and 60% RH, which can be explained as a series of generators'- is wrong. Series connection may increase the voltage not the current. It maybe associated with the enhanced surface charge sites that eventually amplify the current. Please give proper reasoning. The authors did not mention how they have regulated the temperature or humidity? Was it done inside a controlled temperature/humidity chamber? Further the authors have used different temperature/humidity conditions for different sets of experiments which is not good.

7) How the redox reaction of the carbon electrodes is coupled to the streaming current mechanism to dictate the overall device performance? If there are continuous redox reactions, the biproducts in the solution may block the pore network which may eventually suppress the device performance in long-term experiments.

8) Authors stated that the integration of i-TE improves voltage, but if we look into the Fig. 2a and 4b, it seems that there is no significant difference in the output voltage (close to 4 V) for both HCEHG and d-Al₂O₃. So, what is the usefulness of using i-TE material or gelatin matrix here?

9) Charging capacitor (100 μ F) for more than 40 min (Fig. 5a) and then using it to power LEDs for few seconds does not meet any practical applications. What is the super-capacitor value used here (no experimental data is found)?

The practical implementation of wearable devices is at its immature stage as it does not mention several parameters: for e.g., how it will harvest power from the human body? Is it by extracting sweat? If it is so then what is the sweat volume, sweat concentration, etc. (on which the output power is dependent).

10) What is the significance of using the i-TE/gelatin matrix throughout the back surface of d-Al₂O₃? Instead, that could be only integrated to the place near the top electrode which will enhance water evaporation from top (not from the whole surface). This way, the author could reduce material cost as well as the evaporation loss from the other surfaces.

11) The adjective flexible is used for both heat conduction and thermoelectric generator. Although the thermoelectric generator part is self-explanatory, the flexible heat conduction concept is not clear.

12) The authors mention "Repeatedly switching the connection mode between the device and the voltmeter, the voltage signal reverses correspondingly, further confirming the proposed working mechanism" It is quite unclear how does such repeated switching validates the working mechanism

13) "the flexible porous d-Al₂O₃ HG can withstand a large range of bending deformations of 0 to 180° without significant attenuation in the power generation performance" The relation between the drying process and bending needs to be explained properly. How this associated bending leads to attenuation of power generation? Mathematical modelling of the bending is required.

14) The thermoelectric material would reduce the temperature available to the hydrovoltaic generator. How does it enhance the evaporation and power generation?

15) The details of the power and voltage generated under combined mode of operation and its comparison to stand alone mode seem missing.

Reviewer #3 (Remarks to the Author):

The study done by Li et. al. reported a flexible heat conduction effect enhanced hydrovoltaic power generator (HCEHG) by integrating a flexible ionic thermoelectric (i-TE) gelatin material with a porous dual-size Al₂O₃ HG. However, the novelty of the design is limited and cannot reach the high standard of the journal 'Nature communication', Some details in the explanation of mechanisms and testing process are missing. I suggest it major revision and the suggestions are listed in detail as follows:

Line 202, there is a 1 K temperature difference between the d-Al₂O₃ film surfaces of the HCEHG (high) and individual hydrovoltaic device (low), further confirming the contribution of the i-TE gelatin to the hydrovoltaic component (Fig S14). '1K' temperature difference is too small and I am wondering the reality of the difference by considering the error from the testing equipment. Also the author didn't mention which kind of device was used for the test.

Fig.5, the display on the phone is too hard to see, it is suggested the author add this graph in the manuscript or supporting information.

In 'Methods', the author explained the fabrication process of the device, but the test procedure and equipment used was unclear.

Line 200, 'which is attributed to the better thermal conductivity of TE gelatin leading to the transfer of more heat from...', the author mention 'better thermal conductivity', however, by comparing which kind of material with the TE gelatin is unclear, also the thermal conductivity of TE gelatin is not shown, which makes this statement suspectable.

Some import parameters of the TE gelatin is not shown, such as ZT, seebeck coefficient.

Fig. 4h and 4j, ' the performance of stably generating an open-circuit voltage of 6.4 V for a single device remarkably outperforms previously reported hydrovoltaic power generators', does the voltage V_{hoc} includes V_{toc} , if so, the author cannot compare the performance of hydrovoltaic power generator together with a thermoelectric generator with a single hydrvoltaic generator.

Fig. 5c, 'the output voltage of the thermoelectric module increased from 1.5 mV to 4.1 mV, which provides a basis for generating electricity from human body heat. ' the current and powder density of the generator by harvesting human body heat is not shown.

Point by point response

Reviewer #1 (Remarks to the Author):

The authors have successfully introduced a new HCEHG based on the combination of flexible i-TE gelatin and aluminum oxide to overcome the ambient temperature and slow heat replenishment limit. A comparingly high voltage is achieved with a solid application example. However, several questions need to be taken into account to further improve the manuscript quality:

Response: We appreciate the reviewer for the positive comments and important suggestions, which help us to greatly improve the quality of our manuscript. We have revised the manuscript carefully and organized our point-by-point response to these comments as below.

1. Why is the proposed porous Al_2O_3 coated to 60 microns thickness? What happens as the thickness in either the PET layer or the HG layer changes? Will it influence the as-proposed nanochannel steady-state reaching rate or streaming potential? The voltage doesn't vary from 2-10cm, what about <2 or >10?

Response: Thanks for reviewer's suggestion. Here, we proposed porous 60 μm Al_2O_3 layer on flexible PET substrate to keep the balance between the mechanical stability and the power generation performance of the device. As shown in the following Fig. a, when the thickness of Al_2O_3 layer is 100 microns, the internal stress and the stiffness of Al_2O_3 layer will increase and induce cracks, resulting degenerative performance of 0.31 V V_{hoc} and 0.44 μA I_{sc} . When the thickness is 60 microns, the device can maintain stable mechanical and electrical properties under 180° bending strain, shown in Fig b. However, when the thickness is reduced to 25 μm , the number of nano-channels in the Al_2O_3 layer decreases, which also reduces the performance, especially the output current, shown in following Fig c.

Fig. a. Optical images showing a HG device with 100 μm Al_2O_3 layer under relaxed and bending state. **b.** Optical images showing a HG device with 60 μm Al_2O_3 layer under relaxed and bending state.

Fig. c V_{hoc} and I_{sc} of HGs with 25, 60, and 100 μm Al_2O_3 layer respectively.

Moreover, we investigate the output voltage of the HG devices with the width <2 and >10 to evaluate the effect of width on power generation performance. As shown in the following figure, when the width is 1 and 12 cm, the open circuit voltage all maintain at about 3.7 V and the short circuit current increased as the width increase. This result proves the open circuit voltage of HGs doesn't vary with the width when the width larger than 1 cm. The above results and discussion have been added to the revised manuscript on page 6 line 146, Fig S9 and Fig S10.

Fig d Variation of V_{hoc} and I_{hsc} with different width of porous $d\text{-Al}_2\text{O}_3$ hydrovoltaic generator device.

2. Since the layer is considered to be automatically thermal gradient, a vertical thermal mapping should be carried out to prove the stated conclusion.

Response: Thanks for reviewer’s helpful suggestion. A vertical thermal mapping has been carried out according to this advice. As shown in the following figure, a distinct thermal gradient is spontaneously constructed from HG layer to i-TE gelatin to air (from low to high temperature), which proves the heat is transferred from the environment through the i-TE gelatin to HG layer. In other words, the thermoelectric module shows a definite heat conduction enhancement function. The above results and discussion have been added to the revised manuscript on page 10 line 229, Fig S23.

Fig. a. Vertical infrared thermal image of the heat conduction effect enhanced hydrovoltaic power generator.

3. The theoretical principle behind has few innovations, but considering the high

hydrovoltaic value, still a very valuable work. Maybe discuss more about intra-layer and interlayer transportation of the electrons and holes.

Response: Thanks for reviewer's suggestion. After a systematic investigation of the works related to hydrovoltaic generators (*Phys Rev B.*, **2004**, 70(20), 205423.; *Electrochem Commun.*, **2006**, 8(11): 1796-1800.; *ACS Nano.*, **2017**, 11(8), 8421-8428.), combined with our results, we explain the mechanism of our hydrovoltaic power generator as follows. Driven by the evaporation of water, the evaporation of water (Q_{eva}) reaches a balance with the capillary seepage flux (Q_{cap}), and the capillary seepage flows along the Y-axis from bottom to top. The positively charged nanochannels repel positively charged ions (H^+) in the evaporation-driven water flow but allow the negatively charged hydroxyl ions (OH^-) to pass through, inducing a streaming current (I_{sc}) and charge accumulation along the flow that forms a negatively charged hydroxyl ions concentration gradient. The accumulated charge forms a diffusion current ($I_{diffuse}$) in the opposite direction of the I_{sc} due to the coulomb action. At steady state, the I_{sc} and $I_{diffuse}$ will reach dynamic equilibrium, i.e. $|I_{sc}| = |I_{diffuse}|$, and a stable open circuit voltage (V_{hoc}) will be generated.

In the dynamic balance state, in the distance closed to the surface of the Al_2O_3 , negatively charged anion can produce a local coulomb field, which will induce directional migration of carrier in Al_2O_3 (*J Phys Chem Lett.*, **2018**, 9(4): 851-857.; *Chem Phys.*, **2007**, 342 (1-3): 163-172.). As water flows through nanochannels constructed of Al_2O_3 nanoparticles, the migration will occur intra-layer and interlayer of Al_2O_3 . Therefore, the device can continuously generate output current without any redox reaction. We repeatedly switching the connection mode between the device and the voltmeter is to confirm the value and direction of the voltage generated by the device are consistent with our proposed mechanism. Moreover, the highlight of this work lies in a heat conduction effect boosted approach to break through the barrier of restricted ambient temperature and slow heat replenishment in the phase transition of water molecules and realizing the significantly improved performance using a flexible ionic thermoelectric (i-TE) gelatin material to boost the heat conduction between HG and near environment, which is the innovative exploration and effective utilization on

the mechanism.

Fig Schematic depiction of the working mechanism of the evaporation-driven HG.

4. Also, the heat and moisture might not be evenly distributed in all the areas of the working device. For example, the heat and moisture distribution may not be as perfect as a heat plate. Will this lead to a variance in the evaporation rate as well as the h-e mobility?

Response: Thanks for reviewer's suggestion. The device used here is a two-electrode planner multilayers structure with a size of a few centimeters. Within the centimeter size, the humidity and temperature can be considered approximately uniform under normal conditions. In the manuscript, we evaluated the effect of heat and humidity on the device from an overall perspective and found high temperature and low humidity can promote water evaporation, h-e mobility and improve the power generation performance, which is reasonable and meaningful.

As the reviewer pointed out, if we scale up device sizes to larger scales, such as size larger than dozens of centimeters, the effect of local temperature changes does need to be evaluated. From the point of heat and moisture, the above statement also applies to small-sized local areas and local variation will also affects the overall output performance of the device. In order to verify the above point of view, we constructed the scenario by placing a heated electric iron at one end of the device (length \times width: $12 \times 3 \text{ cm}^2$), as shown in the following Figure a-b, to simulate extreme uneven heat

distribution. When the soldering iron temperature is 323.15 K, the open circuit voltage of the device increases slightly from 4.09 V to 4.23 V. Here the room temperature is 306.15 K and relative humidity is 43 %. When the soldering iron temperature is 423.15 K, the open circuit voltage of the device increases from 4.08 V to 4.47 V. These results confirm high temperature and low humidity can improve the power generation performance in the local small size range. The above results and discussion have been added to the revised manuscript on page 7 line 163, Fig S15.

Fig. a. Optical photographs show the test conditions. **b.** Variation of V_{hoc} as a function of time.

5. Fig. 1 is composed of three figures: left, middle, and right. The direction of ammeters in the left and right figures should be the direction of positive charges, and their arrow directions are reversed. The current direction in the right diagram is reversed. The direction of the left picture is confusing. In my point of view, water vapor should move upward. So the left picture should be rotated 90 degrees counterclockwise. In this way, the meters in the left figure and the middle figure are connected at the upper and lower ends, which is easy to understand.

Response: Thanks for reviewer's suggestion. According to the suggestions, we have made detailed modifications to Fig 1, as shown in the following figure.

Fig 1. Schematic diagram of the structure and mechanism of the HCEHG for sustained evaporation electricity output and thermoelectric conversion without any special environmental requirements.

6. Fig S2 needs to add labels to mark these layers.

Response: Thanks for reviewer's suggestion. We have added labels to Fig S2, shown in the following figure.

Fig. S2 Cross-sectional Scanning electron microscope (SEM) images of porous $d\text{-Al}_2\text{O}_3$ film.

7. In lines 110-111, why switching the connection can prove this mechanism. This needs more explanation. In most cases, changing the positive and negative connection of the voltmeter will not change the absolute value measured. Under what circumstances will it change? What would happen if it changed?

Response: Thanks for reviewer's suggestion. Repeatedly switching the connection mode between the device and the voltmeter is to confirm the value and direction of the voltage generated by the device. As shown in the Fig S5, the voltage absolute value didn't change but signal direction reverses correspondingly when switching the connection mode and the upper electrode corresponds to the low potential. These results are consistent with the streaming potential theory proposed in this paper: the positively charged nanochannels repel positively charged ions (H^+) in the evaporation-driven water flow but allow the negatively charged hydroxyl ions (OH^-) to pass through, inducing a streaming potential and charge accumulation along the flow that forms an electric field. As a result, the device reaches a state where the voltage is numerically stable and the top electrode is negative, which is the meaning of switching the connection mode. On the contrary, if the value of the voltage changes but the direction remains the same, the device is not a reliable hydrovoltaic device. We have elaborated further in the revised article on page 5 line 112.

8. The chemical equation in lines 150 and 153 needs to be modified. " FeCN_4^- " should be $\text{Fe}(\text{CN})_6^{4-}$. " FeCN_3^- " should be $\text{Fe}(\text{CN})_6^{3-}$. In line 303, the unit of 60 is missing.

Response: Thanks for reviewer's careful review. We have corrected the typographical errors in the revised manuscript.

9. In Fig 4 f-g, why is the middle area low temperature? If it's convenient, please add the optical image.

Response: Thanks for reviewer's careful review. The low temperature in the middle area is caused by the transfer of heat to the HG layer through a thermoelectric ionic gel with the thermal conductivity of $0.463 \text{ W m}^{-1}\text{K}^{-1}$. The edge area of thermoelectric

module is a circle of Ecoflex with relatively low thermal conductivity (**Fig e.**), and heat cannot be well transmitted through it. Therefore, temperature of the edges is higher than that of the middle region. We remade the device with good interface combination and re-measured its infrared thermal imaging, as shown in the figure a-d. We have updated the corresponding results in the revised manuscript Fig 4 and Fig S20.

Fig a-b. Infrared thermal images of the $d\text{-Al}_2\text{O}_3$ film surface (a) and TG module surface (b) under an optical density of 1 kW m^{-2} . c-d. Optical images showing the HG surface (c) and back side (d) HCEHG.

Fig e. Schematic diagram of the construction of the HCEHG.

10. The comparison of Fig S14 may not be appropriate. Authors can coat the back of Al₂O₃ with black carbon film to harvest solar irradiation.

Response: Thanks for reviewer's suggestion. In this work, we demonstrated a flexible heat conduction effect enhanced hydrovoltaic power generator (HCEHG) by integrating a flexible ionic thermoelectric (i-TE) gelatin material with a porous Al₂O₃ HG. The TE module can improve the heat conduction between the *d*-Al₂O₃ film and the near environment whether there's solar irradiation or not, thus increasing the evaporation rate of water to improve the performance of the HG. Synergistically, the HG module with continuous water evaporation can provide a constant temperature difference for the thermoelectric generator. Fig S14 shows the comparison of the temperature of the devices with and without TE module at indoor condition without solar irradiation to highlight the heat conduction enhancement function of the TE module. Therefore, it's not necessary to coat the back of Al₂O₃ with black carbon film to harvest solar irradiation. We have improved the description in the revised manuscript on page 10 line 225 to emphasize the point.

11. In line 266, it says the neck is gradually bent, but there are no gradually changed

signals in Fig S17 instead of pulsed signals. A pressure sensor should detect pressure. And the pressure should be gradually changed if the neck is gradually bent. If the neck is suddenly turned to one degree and holds the position, the signal should also hold at a high position.

Response: Thanks for reviewer's suggestion. In this work, a flexible CNT pressure sensor powered by the power generation was attached onto the neck. When the neck is bent, the movement of the neck muscle squeezes the pressure sensor, which will trigger the resistance change of the pressure sensor and realizes the monitoring of the neck bending movement. Indeed, as the reviewer suggested, the real-time current change curve in Fig S17 can't show the neck motion very well. Therefore, we conducted additional test to record the motion of the neck during quickly turning to 15°, 30°, 45°, and 60° and holding the position for 10 s. The current change curve is shown in the following figure, which is the reasonable demonstration of the neck movements in real time. The corresponding description and results have been revised in the manuscript on page 13 line 290 and Fig S27.

Fig S27. Real-time current change of the CNT pressure sensor driven by thermoelectricity module showing the bending angle of neck.

Reviewer #2 (Remarks to the Author):

The paper presents ‘a flexible heat conduction effect enhanced hydrovoltaic power

generator (HCEHG) by integrating a flexible ionic thermoelectric (i-TE) gelatin material with a porous dual-size Al₂O₃ (d-Al₂O₃) HG. The i-TE gelatin material can effectively improve the heat conduction between HG and near environment, that increases the water evaporation rate to improve the output voltage of the HPG to 4.0 V at 294.6 K and 30% RH. The HG part with continuous water evaporation can induce a constant temperature difference for the thermoelectric generator (TG). The system can further efficiently achieve solar-to-thermal conversion to raise the temperature difference, accompanied by a stable open circuit voltage of 6.4 V for the HG module.'

Though various experimental observations are performed, the idea of establishing such an evaporation induced power generator is not new. (the parametric results should be included in the ESI and the main findings should be in the main manuscript). In addition, there are several fundamental aspects which the authors did not address properly and therefore cannot be recommended for publication. For example:

Response: Thanks for the reviewer's opinion. Although hydrovoltaic generator (HG) driven by water evaporation has been developed since 2017 (Nat. Nanotech., 2017, 12, 317–321.), restricted ambient temperature and slow heat replenishment in the phase transition of water molecules severely limit the performance of the evaporation-induced hydrovoltaic generators, which is recognized as one of the biggest bottleneck problems of hydrovoltaic field.

Under this background, the novelty of our work can be summarized as follows: Firstly, for the first time, we developed a heat conduction effect boosted approach to break through the above barrier and realize the significantly improved performance using a flexible ionic thermoelectric (i-TE) gelatin material to boost the heat conduction between HG and near environment. Secondly, the ingenious dual-size Al₂O₃ nanoparticle design can well realize the increase of nanochannels and the selectivity improvement. Moreover, the device can efficiently achieve solar-to-thermal

conversion to raise the temperature, accompanied by a stable open circuit voltage of 6.4 V for the HG, the highest value yet comparing to reported HGs. Finally, the HG with continuous water evaporation can induce a constant temperature difference for the thermoelectric generator, providing a promising method for the synergistic harvesting multiple environmental energy sources (thermal, solar, etc.) to generate electricity. The approach proposed in our work is remarkably effective to break through the above limitation and realize more extensive application of HGs.

We appreciate the reviewer for the specific comments and important suggestions. We have refined the fundamental aspects and some details of the manuscript. In addition, the framework of the article has been adjusted. Our point-by-point response to these comments is organized as below.

1) What is the reason for using dual size particles? What is the effect of dual size nanoparticle mass ratio on the capillary filling rate? In connection to this, how the hierarchical nanoparticle structures contribute to the streaming potential/current?

Response: Thanks for the reviewer's opinion. We prepared the *d*-Al₂O₃ film by using two sizes of Al₂O₃ nanoparticles with diameters of 200 nm and 20 nm respectively, mainly because the smaller-sized Al₂O₃ can be filled into the gap that formed by the larger-sized Al₂O₃ particles to form more mechanically stable nanochannels with better ion selectivity. As shown in the following schematic diagrams Fig 1a-b, the embedding of small particles can significantly increase the number of channels in the film, while the size of the channels decreases. At the same time, as the channel size decreases, the overlapping part of the electrical double layer on the channel wall increases, which enhances the selectivity of the channel to ions. That also allows our HG device to operate with easily accessible distilled water (resistivity ~ 0.1 MΩ cm) rather than high purity deionized water (resistivity ~ 18.2 MΩ cm) as reported hydrovoltaic devices (Nat. Nanotech., 2017, 12, 317–321.).

Fig 1a-b schematic diagrams of flow channels in single-size Al_2O_3 film (diameter :200 nm) and $d\text{-Al}_2\text{O}_3$ film.

To investigate the effect of dual size nanoparticle mass ratio on the capillary filling rate, we fabricated a series of $d\text{-Al}_2\text{O}_3$ film with different mass ratios. As shown in the following figure, with the increase of 20 nm Al_2O_3 particles from 1:0 to 5:1 ($m_{200} : m_{20}$, m_{200} and m_{20} are the mass of 200 and 20 nm Al_2O_3 respectively), the capillary filling time increases from 2380 to 2890 s, which can be explained by flow resistance R_h . The flow resistance R_h of a channel with cross-section area A and length l has been widely proven (Sensors and Actuators B. 111–112 (2005) 385–389):

$$R_h = \frac{8\pi\eta l}{A^2}$$

Where η is the viscosity of the solution. The reduced cross-sectional area can increase the R_h .

We then compared the current and voltage signals of HGs with different mass ratios. As shown in the following figure 2a, when the mass ratio of 200 nm Al_2O_3 to 20 nm Al_2O_3 particles is more than 5:1, open circuit voltage doesn't show obviously change. However, the short circuit current of the devices increases from 1.03 to 1.33 μA when the mass ratio of 200 nm Al_2O_3 to 20 nm Al_2O_3 decreases from 1:0 to 50:1. It can be understood that a larger number of channels induced by hierarchical nanostructures at a reasonable scale is helpful for current increases. However, when mass ratio of 20

nm Al₂O₃ particles is 5:1, the open circuit voltage and short circuit current dramatically decrease. This is because, on the one hand, overweight 20 nm Al₂O₃ nanoparticles induced a significant increase in flow resistance; on the other hand, under the same preparation conditions, it was difficult to release the stress in the process of solvent evaporation among overweight 20 nm Al₂O₃ particles, resulting in cracks (shown in the following figure b).

Therefore, we use the hierarchical structure with appropriate proportions in terms of the power generation performance, mechanical stability and adaptability of the final HG device. The above results and discussion has been added to the revised manuscript on page 6 line 136, Fig 2d and Fig S7.

Fig. **a** Variation of filling time, V_{hoc} , and I_{hsc} as a function of mass ratio of 200 and 20 nm Al₂O₃. **b**. optical images showing a HG device with m₂₀₀ : m₂₀ of 5:1 d-Al₂O₃ layer under relaxed state.

2) What kind of pore model is considered for explaining the streaming current mechanism? It seems (from SEM image), the pores are not well defined and randomly distributed, so how does it affect the current—is there any leakage current? There is a high chance of leakage current as only a few fractions of the pores are responsible for generation of streaming current. The streaming current mechanism cannot be explained by simply considering straight nanochannels.

Response: Thanks for the reviewer's suggestion. As shown in the SEM images, the d-Al₂O₃ layer is full of the nanochannels composed of Al₂O₃ nanoparticles, whose

overall diameters are in tens of nanometers. Based on this design, we can build the following structural model, as shown in the following figure. Firstly, as water evaporates from the surface of the film, a pressure difference between the two ends of the Al_2O_3 channel drives water through the channel. Since evaporation occurs on the entire surface of the film, on the one hand, evaporation at the top of the film will drive water flow upward to replenish. Meanwhile, evaporation at the surface of the film will drive water flow upward along the cross section (that is, perpendicular to the film) to replenish. Therefore, the Al_2O_3 constructed nanochannels do not change the overall upward direction of water flow. Besides, according to the test results, the resistance $d\text{-Al}_2\text{O}_3$ nanoparticles formed film after filling with water reaches $2\text{ M}\Omega$, which is very large to further limit the formation of leakage current.

In the traditional calculation of the streaming potential, the streaming current satisfies the following equation:

$$I_s = \frac{A\varepsilon_0\varepsilon_r}{\eta l} \Delta P \zeta$$

where ε_0 , ε_r and η are the vacuum permittivity, relative permittivity of solution and viscosity of the solution, A is the cross-section area, l is the channel length, ΔP is the pressure difference at both ends of the channel. ζ is the Zeta potential. Compared with the above simple straight channel model, our randomly distributed channels only change the geometrical morphology of channels, including diameter and length, while the core mechanism (electrokinetic mechanism under selective overlapping double layers) affecting streaming potential/current is still reasonable.

Fig 1. Schematic diagram of the structure and water flow direction in the *d*-Al₂O₃ film constructed HG.

3) Why is the carbon tape used in this measurement? Generally, Ag/AgCl electrodes are used for such measurements.

The capacitance of the carbon electrode in contact with the solution, will then play a role, because at these low voltages it is less likely that Faradaic reactions occur at the carbon electrodes may lead to a large voltage drop at the electrodes that may cause water electrolysis, complicating the analysis.

Response: Thanks for the reviewer's opinion. Carbon materials are generally considered to be very chemically inactive materials, which are already used in hydrovoltaic devices (Nat. Nanotech., 2017, 12, 317–321; Nano Energy., 2019, 60, 52–60; Adv. Funct. Mater. 2017, 27, 1700551.). Meanwhile, the carbon paste we used here has good electrical conductivity and printability characteristics, which can be readily and controllably printed on a flexible substrate. During the fabricating process of the HG device, the carbon electrode also shows great mechanical property matching with the scraped *d*-Al₂O₃ layer.

Ag/AgCl electrodes is a type of electrode pairs that are prone to redox reactions, which can convert an ion stream into an electron stream by continuous redox reaction between the two electrodes. The electrode reaction can be expressed as the following equation:

In the case of pure water, the Ag/AgCl electrode pair is unstable, and the chloride ions on the electrode will diffuse into the solution, thus affecting the normal operation of the HG device.

4) As the zeta potential of *d*-Al₂O₃ is relatively high (48 mV), a higher streaming current is expected, however, it is observed to be low here. On the other hand, the complete capillary filling time or voltage equilibrium time (Fig. 2a) is found to be quite high (around 41 min) which maybe due to the high channel resistance. Is it the

mass ratio or dual particle size influencing the device performance?

Response: Thanks for the reviewer's opinion. In general, for channels of the same scale, higher Zeta potential helps to improve the ion selectivity of the channel. In fact, in addition to Zeta potential, the performance of the HG device is also affected by the size and number of channels, the wettability of the material, the evaporation rate of water and the device size. For example, the device reported by Wanlin Guo (Nat. Nanotech., 2017, 12, 317–321) with a materials zeta potential of -33.2 mV shows an open circuit voltage of about 1V and short circuit current of 100 nA. While the device reported by Jiming Bian (Nano energy., 2019, 57, 269-278.) with the materials zeta potential of +27.9 mV shows the open circuit voltage of 0.7 V and short circuit current of 1.3 μ A. So, it proves that the device current is not only affected by the surface potential of the material.

In the response to the question 1, we have shown that the reduced cross-sectional area can increase the R_h , resulting a high capillary filling time. But for a HG device that can sustainable to produce power for a long time, the 40 min capillary filling time is negligible.

We then investigate the influencing of mass ratio of the d - Al_2O_3 on the device performance. As shown in the following figure a, when the mass ratio of 200 nm Al_2O_3 to 20 nm Al_2O_3 particles is more than 5:1, open circuit voltage doesn't show obviously change. However, the short circuit current of the devices increases from 1.03 to 1.33 μ A when the mass ratio of 200 nm Al_2O_3 to 20 nm Al_2O_3 decreases from 1:0 to 50:1. It can be understood that a larger number of channels induced by hierarchical nanostructures at a reasonable scale is helpful for current increases. However, when mass ratio of 20 nm Al_2O_3 particles is 5:1, the open circuit voltage and short circuit current dramatically decrease. This is because, on the one hand, overweight 20 nm Al_2O_3 nanoparticles induced a significant increase in flow resistance; on the other hand, under the same preparation conditions, it was difficult to release the stress in the process of solvent evaporation among overweight 20 nm Al_2O_3 particles, resulting in cracks (shown in the following figure b). The above results and discussion have been added to the revised manuscript on page 6 line 136,

Fig 2d and Fig S7.

Fig a. Variation of filling time, V_{hoc} , and I_{hsc} as a function of mass ratio of 200 and 20 nm Al_2O_3 .

5) When there is any bending deformation (Fig. 2d), the nanopores length would increase and that should decrease the output voltage. On the other hand, the increase in the surface area due to bending should increase the surface conduction effect that may lead to reducing the streaming current. On the contrary, the increased surface area also leads to an increase in the evaporation rate which should increase the current. It's quite surprising that there is no change in the output voltage---the author should judiciously justify all the contradictory factors.

Response: Thanks for the reviewer's opinion. To explain the reviewer's doubts, we simulated our device using COMSOL to obtain the strain variation of Al_2O_3 layer during the bending process of the device (following Figure a-b). As shown in the Figures, the maximum strain occurs near the middle of the Al_2O_3 layer and is only 0.111 % when the bending angle is 180° . In other words, the surface area of the part of Al_2O_3 layer with the most severe strain is increased by 0.111%. The above result can be attributed to the thin Al_2O_3 layer (thickness $\sim 60 \mu m$) and the PET substrate (thickness $\sim 200 \mu m$) and large device size (length \times width: $10 \times 3 \text{ cm}^2$). For a 50 nm diameter nanopore, the diameter increase is 0.0555 nm, which is four orders of magnitude smaller compared to the several-hundred-nanometers Debye length of

water. As a whole, the strain of Al_2O_3 layer is significantly less than 0.111%, and the area increase of Al_2O_3 layer ($10 \times 3 \text{ cm}^2$) is much less than 0.0333 cm^2 . Since less than 0.111% strain induced by bending 180° induced changes in nanopores, the surface conduction effect, and surface area is extremely small for a 30 cm^2 device and that is not evident in the signal. Obviously, the strain introduced by $< 180^\circ$ bending angle is smaller. Therefore, the signal of the device does not change significantly during the change of bending angle. The above results and discussion have been added to the revised manuscript on page 6 line 140 and Fig S8.

Fig. **a** Schematic diagram of modeled device model. **b** COMSOL simulation diagram of strain distribution on the device. The bending angle is 180° .

6) The statement – ‘ I_{hsc} increased from 0.32 to 1.34 μA when the width changed from 2 to 10 cm at 303.2 K and 60% RH, which can be explained as a series of generators’- is wrong. Series connection may increase the voltage not the current. It maybe associated with the enhanced surface charge sites that eventually amplify the current. Please give proper reasoning. The authors did not mention how they have regulated the temperature or humidity? Was it done inside a controlled temperature/humidity chamber? Further the authors have used different temperature/humidity conditions for different sets of experiments which is not good.

Response: Thanks for the reviewer’s suggestion. I_{hsc} increases approximately linearly from 0.32 to 1.34 μA when the width changed from 2 to 10 cm at 303.2 K and 60% RH and the generated voltage of the obtained $d\text{-Al}_2\text{O}_3$ HG module does not vary with the width. This result is consistent with the circuit law of multiple parallel devices with the same voltage.

We have revised the manuscript to show the results of a series of devices in parallel. We control temperature and moderation by high and low temperature/humidity test chamber manufactured by Guangzhou wusuo environmental equipment Co., LTD. And the tests were done inside the temperature/humidity test chamber.

As the reviewer pointed out, some of our tests did not use exactly the same temperature/humidity environmental state. That is mainly due to some experiments need to be carried out in an open indoor environment, such as exploring the influence of airflow velocity, etc. And the temperature and humidity parameters of the surrounding environments, where all of our data in this article are tested, are accurately recorded and marked in the manuscript. The relevant expressions have been corrected in the revised article on page 6 line 148.

7) How the redox reaction of the carbon electrodes is coupled to the streaming current mechanism to dictate the overall device performance? If there are continuous redox reactions, the biproducts in the solution may block the pore network which may eventually suppress the device performance in long-term experiments.

Response: Thanks for the reviewer's opinion. Carbon is generally considered to be very chemically inactive material, which are widely used in hydrovoltaic devices (Nat. Nanotech., 2017, 12, 317–321. Nano Energy., 2019, 60, 52–60. Adv. Funct. Mater. 2017, 27, 1700551.) and there's no obvious redox reaction has been reported. To confirm whether the carbon electrode redox reaction occurred in our work, we firstly observed the performance of the HG device when the evaporation gradually stops by sealing the normally working HG device. As shown in the following figure a, the open circuit voltage dropped gradually from 3.68 V and finally died out within 2500 s, indicating a strong correlation between the induced voltage and water evaporation. At the same time, the possibility of electricity generated by electrodes redox reaction is ruled out.

Then we further investigated the X-ray photoelectron spectroscopy (XPS) of C elements of the top and bottom electrodes (the Al₂O₃ covered part) of a device after continuous working for 24 hours. As shown in the following figure a-b, high-resolution C1s XPS spectra of up and bottom electrode are almost the same,

which proves that there is no significant redox reaction of carbon on the top and bottom electrodes. As a matter of fact, in our work, we used carbon as electrodes and the constructed hydrovoltaic devices can operate steadily for a long time (over 7000 s). We have added the above results and discussion to the revised manuscript on page 4 line 101 and Fig S3.

Fig. high-resolution X-ray photoelectron spectroscopy (XPS) of C1s elements of the bottom (a) and top (b) electrodes.

8) Authors stated that the integration of i-TE improves voltage, but if we look into the Fig. 2a and 4b, it seems that there is no significant difference in the output voltage (close to 4 V) for both HCEHG and d-Al₂O₃. So, what is the usefulness of using i-TE material or gelatin matrix here?

Response: Thanks for the reviewer's opinion. Fig. 2a shows a V_{oc} of approximately 4.0 V was achieved by the HG under 296.0 K and 22.3% RH, while Fig. 4b presents the V_{oc} of the single d-Al₂O₃ HG is 3.4 V and the V_{oc} of the HCEHG is 4.0 V at 294.6 K and 30.0% RH. That is, HCEHG with i-TE material can achieve an open-circuit voltage of about 4 V at lower temperatures (lower 1.4 K) and higher humidity (higher 7.7% RH), realizing a significant improvement comparing to the single d-Al₂O₃ HG, which is due to the heat conduction enhancement of the i-TE material.

9) Charging capacitor (100 μ F) for more than 40 min (Fig. 5a) and then using it to power LEDs for few seconds does not meet any practical applications. What is the

super-capacitor value used here (no experimental data is found)?

Response: Thanks for the reviewer's opinion. In this work, we used the HCEHG to charge commercial capacitors with capacitances of 2.2, 10, 47, and 100 μF , and the capacitance values are given by producers (Nantong Jianghai Capacitor Co., Ltd. China). As a result, we were able to charge these capacitors to 5.2V with a single charger, demonstrating the important property of the storage of the energy generated by our HCEHG. This feature means that our device is capable of charging the energy storage with capacity over 100 μF to power for more electrical devices. As a sustainable green environmental energy harvesting device, constant and steady work for a long time is a great advantage over the triboelectric or piezoelectric devices of pulsed electrical signals. Besides, the power consumption of a lot of wearable devices is very low, for example the power consumption of our previous work "e-skin" is about 0.12 μW (Adv. Mater. 2014, 26, 1336–1342), which is far less than the maximum output power of 2.36 μW designed in this work. Our device and the energy storage device charged by HCEHG can meet the above power consumption requirements, further proving its usefulness.

The practical implementation of wearable devices is at its immature stage as it does not mention several parameters: for e.g., how it will harvest power from the human body? Is it by extracting sweat? If it is so then what is the sweat volume, sweat concentration, etc. (on which the output power is dependent).

Response: Thanks for the reviewer's opinion. We designed a wearable power-sensor system to demonstrate real-time detecting physiological signals by integrating the HCEHG, a flexible CNT pressure sensor and a Bluetooth device. In order to ensure the stable operation of the evaporation-driven hydrovoltaic generator, we choose superabsorbent hydrogel as the water source (similar approach which has been demonstrated in our previous work in Nano Energy, 2020, 72, 104663). The component of superabsorbent hydrogel is polyacrylic acid branches, which can absorb an amount of water more than 150 times its mass and release water slowly to supply HG at ambient conditions. The human body is a permanent heat source with a surface

temperature of approximately 32°C. Hence, body heat energy can be harvested and converted into electricity through thermoelectric modules and part of the heat is also transferred to the HG module to improve its performance. The way of our designed wearable power-sensor system works is not based on sweat. The above results and discussion have been added to the revised manuscript on page 12 line 276.

10) What is the significance of using the i-TE/gelatin matrix throughout the back surface of $d\text{-Al}_2\text{O}_3$? Instead, that could be only integrated to the place near the top electrode which will enhance water evaporation from top (not from the whole surface). This way, the author could reduce material cost as well as the evaporation loss from the other surfaces.

Response: Thanks for the reviewer's opinion. In this work, we fabricated a flexible heat conduction effect enhanced hydrovoltaic power generator (HCEHG) by rationally integrating flexible ionic thermoelectric (i-TE) gelatin on the back of a porous $d\text{-Al}_2\text{O}_3$ constructed a novel HG as heat conduction layer. The significance of using the i-TE/gelatin matrix is as follows: (1) The i-TE material can improve the heat conduction between the $d\text{-Al}_2\text{O}_3$ film and the near environment, thus increasing the evaporation rate of water to improve the performance of the HG. (2) Harvesting energy from the constant temperature difference induced by HG with continuous water evaporation.

In order to validate the reviewer's suggestion, we only integrated a 5 mm wide i-TE/gelatin matrix near the upper electrode and then tested the performance of the device. As shown in the following figure a, the V_{hoc} and V_{toc} are 3.6 V and 0.8 mV respectively, which are all smaller than that of HCEHG with full coverage of i-TE/gelatin matrix. That can be reasonably explained. As shown in the following figure b, the $d\text{-Al}_2\text{O}_3$ porous film is an open system and the pores in the $d\text{-Al}_2\text{O}_3$ porous film are countless and partially connected. Due to the water evaporation, there is water flow perpendicular to the film direction beside the water flow along the film direction, which has an impact on the power generation performance of the device. When fully covered by i-TE/gelatin matrix, water flow can be accelerated in both directions simultaneously. However, when the thermoelectric module is only near the

top electrode, the i-TE module has no effect on promoting the water flow in the vertical direction on the film. Moreover, the weakening effect of the flow resistance of the nanochannel makes the performance improvement of HG even smaller. Therefore, only integrating i-TE/gelatin matrix to the place near the top electrode is impossible to achieve the maximum enhancement of device performance.

Fig. a. V_{hoc} and V_{toc} of the devices with TE module covered near top electrode and full area. b. Schematic diagram of the structure and water flow direction in the d -Al₂O₃ film constructed HG.

11) The adjective flexible is used for both heat conduction and thermoelectric generator. Although the thermoelectric generator part is self-explanatory, the flexible heat conduction concept is not clear.

Response: Thanks for reviewer’s suggestion. We have revised the description in the revised manuscript. In the manuscript, we demonstrated a flexible heat conduction effect enhanced hydrovoltaic power generator (HCEHG) by integrating a flexible ionic thermoelectric (i-TE) gelatin material with a porous dual-size Al₂O₃ (d -Al₂O₃) HG. Here, we want to express that a flexible hydrovoltaic power generator with the feature of heat conduction effect enhancement. In terms of the form of the device, all the components including thermoelectric materials, encapsulation, PET and HG are flexible.

12) The authors mention “Repeatedly switching the connection mode between the device and the voltmeter, the voltage signal reverses correspondingly, further

confirming the proposed working mechanism” It is quite unclear how does such repeated switching validates the working mechanism

Response: Thanks for reviewer’s suggestion. Repeatedly switching the connection mode between the device and the voltmeter is to confirm the value and direction of the voltage generated by the device. As shown in the Fig S5, the voltage absolute value didn’t change but signal direction reverses correspondingly when switching the connection mode and the upper electrode corresponds to the low potential. These results are consistent with the streaming potential theory proposed in this paper: the positively charged nanochannels repel positively charged ions (H^+) in the evaporation-driven water flow but allow the negatively charged hydroxyl ions (OH^-) to pass through, inducing a streaming potential and charge accumulation along the flow that forms an electric field. The accumulated charge forms a diffusion current ($I_{diffuse}$) in the opposite direction of the I_{sc} due to the coulomb action. At steady state, the I_{sc} and $I_{diffuse}$ will reach dynamic equilibrium, i.e. $|I_{sc}| = |I_{diffuse}|$, and a stable open circuit voltage (V_{hoc}) will be generated. As a result, the device reaches a state where the voltage is numerically stable and the top electrode is negative. That is the meaning of switching the connection mode. On the contrary, if the value of the voltage changes but the direction remains the same, the device is not a reliable hydrovoltaic device. The relevant description have been corrected in the revised article on page 5 line 112.

Fig Schematic depiction of the working mechanism of the evaporation-driven HG.

13) “the flexible porous $d\text{-Al}_2\text{O}_3$ HG can withstand a large range of bending deformations of 0 to 180° without significant attenuation in the power generation performance” The relation between the drying process and bending needs to be explained properly. How this associated bending leads to attenuation of power generation? Mathematical modelling of the bending is required.

Response: Thanks for reviewer’s suggestion. The flexible porous $d\text{-Al}_2\text{O}_3$ HG is obtained after scrape coating the $d\text{-Al}_2\text{O}_3$ slurry on the PET substrate with electrodes and leaving to dry at room temperature. The bending process is to bend the dry HG device at a certain angle, then place the bottom of the device below the water surface to make it work and investigate its performance. The bending angle of the device is precisely regulated by controlling the chord length d , shown in the following figure. The relationship between chord length d and bending angle θ and arc length l (i.e., HG length 12 cm) satisfies formula 1-2.

$$l \cdot \frac{360^\circ}{\theta} = 2\pi r \dots\dots\dots (1)$$

$$d = 2r \sin \frac{\theta}{2} \dots\dots\dots (2)$$

Fig. Schematic diagram of device bending model.

Such research can display the stable mechanical structure of HGs (no cracks to degrade HG performance), meanwhile, it can also prove the stability of the power generation performance. Due to the thin Al_2O_3 layer (thickness $\sim 60 \mu\text{m}$) and the PET substrate (thickness $\sim 200 \mu\text{m}$) and large device size (12 cm), the changes in geometry induced by bending are extremely subtle. We modeled our device and simulated it

using COMSOL to obtain the strain variation of Al₂O₃ layer during the bending process. The specific settings are in the response to question 5. When bended to 90° and 180°, the maximum strain values of the Al₂O₃ layer are 0.0539% and 0.111% respectively. Such changes are extremely small and hardly reflected in our millivolt level of accuracy tests. Therefore, we declared that “the flexible porous *d*-Al₂O₃ HG can withstand a large range of bending deformations of 0 to 180° without significant attenuation in the power generation performance”. The relevant results and descriptions have been added to the revised manuscript on page 6 line 140 and Fig S8.

Fig. a-b. COMSOL simulation diagram of strain distribution on the device. The bending angle is 90° (a) and 180° (b).

14) The thermoelectric material would reduce the temperature available to the hydrovoltaic generator. How does it enhance the evaporation and power generation?

Response: Thanks for reviewer’s suggestion. The fundamental equation of heat conduction of Fourier's law can be expressed as:

$$q = -k \nabla T \quad (1)$$

$$\frac{dQ}{dt} = -kA \nabla T \quad (2)$$

where q is the heat flux per unit area, k is the thermal conductivity of the material, ∇T is temperature gradient, dQ/dt is the heat conductivity of an object per unit time and A is area of thermal conductivity. This clearly shows that increasing the thermal conductivity and area of the material can effectively increase heat transfer. For our device, the thermal conductivity of the TE gelatin used here is 0.463 W m⁻¹K⁻¹, which is more than 17 times of normal indoor air (~0.0267 W m⁻¹K⁻¹). Besides, the larger

gelatin surface can also increase the heat conduction area, shown in the following figure a. Therefore, from a theoretical point of view, the working principle of our device is feasible. The infrared thermal images indicated the temperature of the HG with thermoelectric module is indeed higher than that without thermoelectric module (following figure 2a-b). The results of our tests showed a significant improvement in device performance from 3.4 V to 4.0 V with the addition of thermoelectric materials, further confirming that our design is sound.

Fig 1a. Schematic diagram showing the heat conduction from environment to device.

Fig 2a-b. Infrared thermal images of the d - Al_2O_3 film surface with and without TG module surface.

15) The details of the power and voltage generated under combined mode of operation and its comparison to stand alone mode seem missing.

Response: Thanks for reviewer's suggestion. We investigated the voltage and effective output power of the device with or without the thermoelectric module state. As shown in the following figure a, the i-TE gelatin part was found to significantly improve the performance of the HG from 3.4 V to 4.0 V at 294.6 K and 30% RH.

Correspondingly, the maximum effective output power increases from 1.84 μW to 2.36 μW at a load resistance of 2 $\text{M}\Omega$. The results represent a significant improvement in device performance. The above results and discussion have been added to the revised manuscript on page 9 line 220 and Fig S21.

Fig. a. Open-circuit voltage response versus time curve of the hydrovoltaic module with and without the thermoelectric module at an ambient temperature of 294.6 K and a humidity of 30% RH. b. Output power of the power generators with and without the thermoelectric module (size: width: 10 cm, height: 3 cm) as functions of the load resistance in a room environment.

Reviewer #3 (Remarks to the Author):

The study done by Li et. al. reported a flexible heat conduction effect enhanced hydrovoltaic power generator (HCEHG) by integrating a flexible ionic thermoelectric (i-TE) gelatin material with a porous dual-size Al_2O_3 HG. However, the novelty of the design is limited and cannot reach the high standard of the journal 'Nature communication', Some details in the explanation of mechanisms and testing process are missing. I suggest it major revision and the suggestions are listed in detail as follows:

Response: Thanks for the reviewer's opinion. Thanks for the reviewer's opinion. Although hydrovoltaic generator (HG) driven by water evaporation has been developed since 2017 (Nat. Nanotech., 2017, 12, 317–321.), restricted ambient

temperature and slow heat replenishment in the phase transition of water molecules severely limit the performance of the evaporation-induced hydrovoltaic generators, which is recognized as one of the biggest bottleneck problems of hydrovoltaic field.

Under this background, we proposed our work, whose highlights can be summarized as follows: Firstly, for the first time, we developed a heat conduction effect boosted approach to break through the above barrier and realize the significantly improved performance using a flexible ionic thermoelectric (i-TE) gelatin material to boost the heat conduction between HG and near environment. Secondly, the ingenious dual-size Al_2O_3 nanoparticle design can well realize the increase of nanochannels and the selectivity improvement. Moreover, the device can efficiently achieve solar-to-thermal conversion to raise the temperature, accompanied by a stable open circuit voltage of 6.4 V for the HG, the highest value yet comparing to reported HGs. Finally, the HG with continuous water evaporation can induce a constant temperature difference for the thermoelectric generator, providing a promising method for the synergistic harvesting multiple environmental energy sources (thermal, solar, etc.) to generate electricity. The approach proposed in our work is remarkably effective to break through the above limitation and realize more extensive application of HGs.

We appreciate the reviewer for the specific comments and important suggestions. We have refined the fundamental aspects and some details of the manuscript. In addition, the framework of the article has been adjusted. Our point-by-point response to these comments is organized as below.

Line 202, there is a 1 K temperature difference between the d- Al_2O_3 film surfaces of the HCEHG (high) and individual hydrovoltaic device (low), further confirming the contribution of the i-TE gelatin to the hydrovoltaic component (Fig S14). ‘1K’ temperature difference is too small and I am wondering the reality of the difference by considering the error from the testing equipment. Also, the author didn’t mention which kind of device was used for the test.

Response: Thanks for reviewer’s suggestion. The temperature data was obtained after

repeated tests using a Fluke TI400 infrared thermal imager with a thermal sensitivity of 0.075 K. To make a strong case for the existence of temperature differences, a vertical thermal mapping has been carried out. As shown in the following figure, a temperature gradient of about 1K was sensitively captured by the Fluke TI400 infrared thermal imager, and the temperature of the Al_2O_3 surface and the black back surface were 19.0 and 20.1 °C, respectively. Here the environment temperature is about 21.5 °C. Based on the performance of the equipment and the specification of our tests, we are confident that the 1 K temperature difference in Fig S14 is reliable and accurate. The above results and discussion have been added to the revised manuscript on page 10 line 22, Fig S228.

Fig. a. Vertical infrared thermal image of the heat conduction effect enhanced hydrovoltaic power generator.

Fig.5, the display on the phone is too hard to see, it is suggested the author add this graph in the manuscript or supporting information.

Response: Thanks for reviewer's kind suggestion. According to your suggestion, we have reedited and added higher definition image in Fig 5.

Fig. Photograph showing the testing process and graphical display by a mobile phone.

In ‘Methods’, the author explained the fabrication process of the device, but the test procedure and equipment used was unclear.

Response: Thanks for reviewer’s suggestion. We have added detailed the test procedure and equipment information in the method.

Characterization:

The Fluke TI400 infrared thermal imager was used to detect the temperature and infrared thermal images. A digital camera (Canon EOS 70D) was used to take the optical photographs and videos in this study. The morphology and element analyses of the samples were examined by scanning electron microscopy (SEM, JSM-7001F). X-ray photoelectron spectroscopy (XPS) data were recorded on an ESCALAB 250 photoelectron spectrometer (Thermo-Fisher Scientific) with Al K α radiation (1486.6 eV). The ambient temperature and humidity were controlled by a temperature and humidity cabinet manufactured by Guangzhou Wusuo Environmental Equipment Co., Ltd. The voltage and current signals were recorded in real time using a Keithley 6514 electrometer, which was controlled by a LabView-based data acquisition system.

Line 200, ‘which is attributed to the better thermal conductivity of TE gelatin leading to the transfer of more heat from...’, the author mention ‘better thermal conductivity’, however, by comparing which kind of material with the TE gelatin is unclear, also the

thermal conductivity of TE gelatin is not shown, which makes this statement suspectable.

Response: Thanks for reviewer's suggestion. The thermal conductivity of the TE gelatin used here is $0.463 \text{ W m}^{-1}\text{K}^{-1}$, which is more than 17 times of normal indoor air ($\sim 0.0267 \text{ W m}^{-1}\text{K}^{-1}$). Therefore, we want to claimed the TE gelatin with better thermal conductivity than air. We have added the above parameters and made modifications and improvements in the revised manuscript on page 3 line 81.

Some import parameters of the TE gelatin is not shown, such as ZT, seebeck coefficient.

Response: Thanks for reviewer's suggestion. We tested and calculated the ZT and seebeck coefficient.

In general, the maximum efficiency of a thermoelectric material for both power generation and cooling was determined by its figure of merit (zT):

$$zT = \frac{(S_e)^2 \sigma}{k} T$$

where S_e is the above mentioned relative Seebeck coefficient, T is absolute temperature, σ is the electrical conductivity, k is the thermal conductivity.

In our TG system, the Seebeck coefficient is defined as:

$$S_e = \Delta E / \Delta T = \Delta S / nF,$$

where ΔE is the open-circuit voltage, ΔT is the temperature difference, n is the number of electrons transferred in the redox reaction, F is Faraday's constant, and ΔS is the partial molar entropy difference of the redox couple. The seebeck coefficient of our TG module was approximately 0.82 mV K^{-1} in the ΔT range of 0.8 to 42.2 K. zT value of our TG module is 0.031 at 296.15 K. The relevant results and discussion have been added to the revised manuscript on page 9 line 192.

Fig. S18. Open-circuit voltage variation of the temperature difference.

Fig. 4h and 4j, ‘ the performance of stably generating an open-circuit voltage of 6.4 V for a single device remarkably outperforms previously reported hydrovoltaic power generators’, does the voltage V_{hoc} includes V_{toc} , if so, the author cannot compare the performance of hydrovoltaic power generator together with a thermoelectric generator with a single hydrovoltaic generator.

Response: Thanks for reviewer’s suggestion. In the manuscript, V_{hoc} represents the open circuit voltage generated by the hydrovoltaic generator (HG) module and V_{toc} represents the open circuit voltage generated by the thermoelectric module. Therefore, V_{hoc} does not include V_{toc} and Fig. 4j is just the performance comparison of hydrovoltaic generator module with the reported single hydrovoltaic generator, which is reasonable.

Fig. 5c, ‘the output voltage of the thermoelectric module increased from 1.5 mV to 4.1 mV, which provides a basis for generating electricity from human body heat.’ the current and power density of the generator by harvesting human body heat is not shown.

Response: Thanks for reviewer’s suggestion. We have tested the current and the power density of the generator by harvesting human body heat. As shown in the following Figure **a-b**, when attached onto human body, the current increased from 65.3 μ A to 232.6 μ A, and the power density of the generator by harvesting human

body heat is 1.1 mW m^{-2} . The above results and discussion have been added to the revised manuscript on page 12 line 20, Fig S26.

Fig a-b. the current and the power density of the generator by harvesting human body heat.

REVIEWER COMMENTS

Reviewer #1 (Remarks to the Author):

The authors have done a careful job in addressing the comments raised by the reviewer. The manuscript is now well-organized. The authors comprehensively investigate the high-performance HCEHG. Nevertheless, based on the revised manuscript, there are three remaining comments that need to be considered before the manuscript is accepted for publication.

1. In Fig. S10, the author states that the cracks are induced by the internal stress and the stiffness of Al₂O₃ layer. And, in Fig. S10, the cracked layer is bent. Is the crack caused by bending or will it be formed without bending in the manufacturing process? If the layer cracks under a relaxed state, the authors may try to plate Al₂O₃ film two times. Leave enough time between the two depositions to relax the stress.
2. The authors use the Seebeck coefficient to describe the voltage is relevant to the temperature. In Fig. S9, the voltage is almost the same while the current keeps increasing with the increase in width. The author could also vary the height of the devices to study whether the voltage/current will change. The authors could provide some data similar to Fig S9 but using the height as the x-axis. The reviewer thinks that there is an upper limit for the current/voltage as the height of the devices increases. In other words, the current/voltage will not increase after the height reaches a specific point. Because the water is only at the bottom of the generator, the temperature difference should decrease as the distance from the water increases. When the height is high enough, the effect of water on the hydropower generator will be weakened. If conditions permit, the authors can vary the heights to study this phenomenon.
3. For Fig. 2e, what is the influence of bending on current? The authors may need to measure the current under different bending angles. Bending will increase the stress in the membrane, which may affect the transportation of ions, and may have a greater impact on the current. However, the authors provided the simulation results. The results showed that the strain change was only 0.111%, which means that the current changes may also be tiny.

Reviewer #2 (Remarks to the Author):

I am satisfied with the revisions made.

point by point response

Editor:

Dear Professor Zhang,

Thank you again for submitting your revised manuscript "Heat conduction effect enhanced hydrovoltaic power generator" to Nature Communications. We have now received reports from two of the reviewers who evaluated the original version. On the basis of their comments (copied below), we have decided to invite an additional revision of your work.

On balance, the reviewers find that your revisions improved the manuscript yet some important points remain to be addressed. You will see that Rev #2 is now satisfied with the revisions of the manuscript. Regretfully, Rev #3 was unable to supply comments following your revision and additional insight was sought from Rev #1. Based on that additional feedback, we ask that you provide some calculations or estimations to support your assertion that a 1 K temperature difference is to generate the output metrics of the device (e.g. the power, voltage, or current) using the reported material. Finally, Rev #1 has also raised some specific additional points additional points that must be addressed in a revised version of your manuscript.

Please revise your manuscript, addressing all the remaining issues raised by the reviewers. If it is not suitably revised, we will be unlikely to send it back to external Review.

Response: Thanks for the editor's suggestions, which is very important for us to improve the quality of our manuscript. In our manuscript, we developed a heat conduction effect boosted approach to break through the barrier of hydrovoltaic generators (HG) and realize the significantly improved performance using a flexible ionic thermoelectric (i-TE) gelatin material to enhance the heat conduction between HG and near environment.

For the i-TE module, to prove that a 1 K temperature difference is suitable enough to generate the output metrics of the device, we calculated the heat-to-electrical

conversion efficiency (η) relative to Carnot efficiency for the i-TE device. For our thermoelectric generator composed of gelatin and $\text{K}_4\text{Fe}(\text{CN})_6/\text{K}_3\text{Fe}(\text{CN})_6$, the heat-to-electrical conversion efficiency relative to Carnot efficiency limit of a heat engine can be expressed by the following equation (*Nat Commun.* **9**:5146 (2018)):

$$\eta_r = \frac{\eta}{(\Delta T/T_H)} = \frac{P_{max}/(\kappa \cdot \frac{\Delta T}{d})}{(\Delta T/T_H)} = \frac{\left(\frac{P_{max}}{(\Delta T)^2}\right)d \cdot T_H}{\kappa} \quad (1)$$

where the electrode separation distance (d) is 3 mm, P_{max} is the maximum power density of 0.196 mW m^{-2} (Fig S24), the hot-side temperature (T_H) is 294.6 K, and κ is the thermal conductivity of the i-TE material, which is measured to be $0.463 \text{ W m}^{-1}\text{K}^{-1}$ by TC3000 thermal conductivity meter fabricated by Xi'an Xiotech electronics Co., Ltd. When temperature difference (ΔT) is 1 K, the η_r is calculated as 0.0374%, which is a very reasonable value comparing with the related thermoelectric works reported recently (for example η_r is 0.01% in *Han et al., Science* **368**, 1091–1098 (2020); 0.79% in *Duan, J. J. et al., Nat Commun.* **9**:5146 (2018)). Therefore, for the thermoelectric part, 1 K temperature difference is suitable enough to generate the current electrical output.

For the hydrovoltaic generator module, enhanced heat conduction effect is the key factor. According to Fourier's law of heat conduction:

$$\Phi = -kA\Delta T \quad (2)$$

where Φ is heat flux, k is thermal conductivity of $0.463 \text{ W m}^{-1}\text{K}^{-1}$, and A is heat transfer area of 30 cm^2 (Length \times width: $10 \text{ cm} \times 3 \text{ cm}$). When ΔT is 1 K, the Φ of our device from thermal module to HG module is 1.389 mW. For the HG module in the steady state, the heat is used for water evaporation and electricity conversion. In the system, 1 K temperature difference induces 1.8 mg h^{-1} incremental evaporation of water, which consumes 1.22 mW thermal energy (ΔP_{eva} , $\Delta P_{eva} = m r$, where m is the quality of the water, r is the latent heat of evaporation of water at the corresponding temperature and pressure). According to Fig S24, the output power of the HG with thermoelectric module is increased by $0.52 \text{ }\mu\text{W}$ compared with that of the independent HG device, which is less than the value of $169 \text{ }\mu\text{W}$ ($\Phi - \Delta P_{eva} = 1389$

$\mu\text{W} - 1220 \mu\text{W}$). From the point of view of energy conservation, the above process can be supported.

Based on the above calculation and discussion, we can determine that a 1 K temperature difference can generate the output metrics of the device from the perspective of energy conversion. Related discussions and calculations have been added to the improved supporting information.

Reviewer #1 (Remarks to the Author)

The authors have done a careful job in addressing the comments raised by the reviewer. The manuscript is now well-organized. The authors comprehensively investigate the high-performance HCEHG. Nevertheless, based on the revised manuscript, there are three remaining comments that need to be considered before the manuscript is accepted for publication.

Response: Thanks for the reviewer's important suggestions, which help us to greatly improve the quality of our manuscript. We have revised the manuscript carefully and organized our point-by-point response to these comments as below.

1. In Fig. S10, the author states that the cracks are induced by the internal stress and the stiffness of Al_2O_3 layer. And, in Fig. S10, the cracked layer is bent. Is the crack caused by bending or will it be formed without bending in the manufacturing process? If the layer cracks under a relaxed state, the authors may try to plate Al_2O_3 film two times. Leave enough time between the two depositions to relax the stress.

Response: Thanks for the reviewer's suggestion. We adopted the reviewer's suggestion to prepare the device through two times scratch coats, and leave 4 h between the two depositions to relax the stress. However, the film, which has been

sufficiently dried for the first time, will crack significantly when exposed to the slurry solvent for the second time (following Figure 1). As the result, there were more obvious cracks on the film in the manufacturing process. For the 100 μm film composed of Al_2O_3 nanoparticles, it is difficult to achieve no cracks by multiple scraping method.

Fig 1. Optical images showing a HG device with 100 μm Al_2O_3 layer fabricated by by two times scraping method.

In figure S10, we displayed cracks on the HG device with 100 μm Al_2O_3 layer under bending state using the optical image. In fact, all of our devices are tested after scrape coating for a long enough time (over 4 h) to allow the solvent (ethanol) to completely volatilize and at this time the device reaches a stable state. After we repeated the experiment many times, we confirmed that cracks would occur without bending in the manufacturing process for 100 μm thick films (following Figure 2a), which is because the rapid evaporation of the solvent (ethanol) will cause the shrinkage of the film, and the thick film cannot release the shrinkage stress sufficiently, resulting in cracks. When we bended the device, the spacing and number of the cracks dramatically increased (following Figure 2b).

Fig 2. a-b. Optical images showing a HG device with 100 μm Al_2O_3 layer under relaxed and bending state.

In general, the relationship between the bending stiffness (EI) and thickness of materials satisfies the following equation:

$$EI = E_{Pa}bh \left(\frac{1}{3}h^2 - hy + y^2 \right)$$

where E_{Pa} represents the Young's modulus, and b , h , and y represent film width, its thickness, and thickness to the natural axis respectively. Because EI exhibits a cubic dependence on the film thickness, the deformation of the device will cause dramatically increase of cracks. Therefore, the mechanical stability of the 100 μm Al_2O_3 layer on flexible PET substrate is not able to meet our requirements for a stable flexible hydrovoltaic power generator. We have improved and enhanced the corresponding description in the article. The above results and discussion have been added to the revised manuscript on page 6 line 145, Fig S13.

2. The authors use the Seebeck coefficient to describe the voltage is relevant to the temperature. In Fig. S9, the voltage is almost the same while the current keeps increasing with the increase in width. The author could also vary the height of the devices to study whether the voltage/current will change. The authors could provide some data similar to Fig S9 but using the height as the x-axis. The reviewer thinks that there is an upper limit for the current/voltage as the height of the devices increases. In other words, the current/voltage will not increase after the height reaches a specific point. Because the water is only at the bottom of the generator, the

temperature difference should decrease as the distance from the water increases. When the height is high enough, the effect of water on the hydropower generator will be weakened. If conditions permit, the authors can vary the heights to study this phenomenon.

Response: Thanks for the reviewer’s meaningful suggestion. We fabricated a series of HG devices with constant width of 4.5 cm and different heights of 1, 2, 3, 4, and 5 cm to investigate the effect of height on the performance. Here, to express the results more clearly and conveniently, we define the open circuit voltage and short circuit current of the device at height x cm ($x = 1, 2, 3, 4,$ and 5 cm) as V_{oc-hx} , and I_{sc-hx} , respectively. Just as the reviewer expected, when the height of the HGs increased from 1 to 4 cm, V_{oc} increased gradually from 0.33 V to 3.97 V, and V_{oc-h3} is 3.82 V. However, V_{oc-h5} is dramatically decreased to 3.04 V (following figure a). Correspondingly, I_{sc} increased from 0.213 to 0.597 μA when height of the HGs increased from 1 to 3 cm. And when height of the HGs is larger than 3 cm, I_{sc} will dramatically decreased.

Fig 1. Variation of open-circuit voltage and short-circuit current with heights of porous $d\text{-Al}_2\text{O}_3$ hydrovoltaic generator device. Here, the width HG devices with constant width of 4.5 cm.

In order to explain the above results, we observed the distribution of water on the Al_2O_3 film. As shown in the following figure, water can climb to a height of 3.5 cm in the Al_2O_3 film at 48% RH, and is distributed in a gradient in the film due to gravity. When height of the HGs increased from 1 to 4 cm, the amount of water flowing

through the nanochannel in the film increases correspondingly, resulting in a higher voltage. However, Al_2O_3 is a very poor conductivity material. When the height of the device is greater than 3.5 cm, the internal resistance of the device will increase sharply, resulting in a rapid decrease in the current of the device. The voltage measured is 3.04 V for the device of 5cm, which is because extra-large internal resistance can also hinder voltage testing on equipment according to the principle of voltage division. The above results and discussion have been added to the revised manuscript on page 6 line 143, Fig S11-12.

Fig 2. a-c. Optical images showing a HG device with the height of 1, 3, and 5 cm.

3. For Fig. 2e, what is the influence of bending on current? The authors may need to measure the current under different bending angles. Bending will increase the stress in the membrane, which may affect the transportation of ions, and may have a greater impact on the current. However, the authors provided the simulation results. The results showed that the strain change was only 0.111%, which means that the current changes may also be tiny.

Response: Thanks for the reviewer's suggestion. In order to evaluate the influence of bending on device performance more comprehensively, we further investigated the current under different bending angles. As shown in the following figure, when the bending angle increased from 0° to 180° , the short circuit current decreased only 0.05 μA from 1.27 μA to 1.22 μA . Therefore, the small strain introduced by bending shows very little the effect on the transportation of ions on the film and the performance of HG device. The above results and discussion have been added to the revised manuscript on page 6 line 135, Fig S8.

Fig. Variation of short-circuit current of the device with 10 cm width at 48% RH.

REVIEWERS' COMMENTS

Reviewer #1 (Remarks to the Author):

The authors have done a careful job in addressing all of the comments raised by the reviewer.

point by point response

REVIEWERS' COMMENTS

Reviewer #1 (Remarks to the Author):

The authors have done a careful job in addressing all of the comments raised by the reviewer.

Response: Many thanks to reviewer for your help for improving our work and recognition of our work.